# Genetics and RNA Regulation of Uveal Melanoma

**DOI:** 10.3390/cancers15030775

**Published:** 2023-01-26

**Authors:** Cristina Barbagallo, Michele Stella, Giuseppe Broggi, Andrea Russo, Rosario Caltabiano, Marco Ragusa

**Affiliations:** 1Department of Biomedical and Biotechnological Sciences—Section of Biology and Genetics, University of Catania, 95123 Catania, Italy; 2Department of Medical, Surgical Sciences and Advanced Technologies G.F. Ingrassia—Section of Anatomic Pathology, University of Catania, 95123 Catania, Italy; 3Department of Ophthalmology, University of Catania, 95123 Catania, Italy

**Keywords:** melanoma, driver mutations, miRNA, mRNA, lncRNA, circRNA, cancer, eye

## Abstract

**Simple Summary:**

Uveal melanoma (UM) is a rare eye cancer with a high mortality rate due to metastases, leading to death in up to 50% of patients within 10 years from UM diagnosis. Moreover, patients show a median survival of 6 to 12 months after metastasis diagnosis. UM and cutaneous melanoma (CM) have the same melanocytic origin; however, they are very different in terms of molecular alterations and biological behavior. In this review, we will discuss the complex genetic and non-coding RNA-based epigenetic landscapes underlying the transformation, progression, and dissemination of UM. This knowledge will pave the way for the future identification of new biomarkers of the pathology and therapeutic targets.

**Abstract:**

Uveal melanoma (UM) is the most common intraocular malignant tumor and the most frequent melanoma not affecting the skin. While the rate of UM occurrence is relatively low, about 50% of patients develop metastasis, primarily to the liver, with lethal outcome despite medical treatment. Notwithstanding that UM etiopathogenesis is still under investigation, a set of known mutations and chromosomal aberrations are associated with its pathogenesis and have a relevant prognostic value. The most frequently mutated genes are *BAP1, EIF1AX, GNA11, GNAQ*, and *SF3B1*, with mutually exclusive mutations occurring in *GNAQ* and *GNA11*, and almost mutually exclusive ones in *BAP1* and *SF3B1*, and *BAP1* and *EIF1AX*. Among chromosomal aberrations, monosomy of chromosome 3 is the most frequent, followed by gain of chromosome 8q, and full or partial loss of chromosomes 1 and 6. In addition, epigenetic mechanisms regulated by non-coding RNAs (ncRNA), namely microRNAs and long non-coding RNAs, have also been investigated. Several papers investigating the role of ncRNAs in UM have reported that their dysregulated expression affects cancer-related processes in both in vitro and in vivo models. This review will summarize current findings about genetic mutations, chromosomal aberrations, and ncRNA dysregulation establishing UM biology.

## 1. Introduction

Uveal melanoma (UM) is the most common primary intraocular malignancy in adults and the most common non-cutaneous melanoma [1]. UM accounts for about 3–5% of all melanoma cases; it develops in the uveal tract of the eye, primarily involving the choroid (85–90%) but also the ciliary body (5–8%) and the iris (3–5%) [2]. The average age of diagnosis is about 60 years; however, UM has been reported in patients of all ages, with younger patients under the age of 18 accounting for only 1% of all cases. UM incidence rises with age, peaking at 70 years old. In children, UM is uncommon, and congenital diseases are extremely sporadic [3]. The prevalence of UM also appears to be gender related [3,4]. Population-based studies have shown that males have a higher age-adjusted incidence than females, with a 20–30% higher rate in males [5,6]. The prevalence of UM is also related to ethnicity. In Europe, the incidence of UM decreases along a north-south gradient due to increased ocular pigmentation in southern populations, which protects the eyes from UV radiation [7]. In 2007, the European incidence ranged from 2 cases per million in southern Italy and Spain to >8 cases per million in Scandinavian countries, with a Central European incidence of 6 cases per million [7]. During the same time period, the incidence in the United States was estimated to be 4.3 cases per million, a value that rises with latitude [8,9]. The average age at diagnosis varies by population, with Asian patients having a lower average.

While the rate of occurrence is relatively low, UM has a high mortality rate due to metastatic spread, with up to 50% of patients dying within 10 years from diagnosis [10,11]; indeed, a median survival of 6 to 12 months after metastasis diagnosis has been reported [1,10]. Because of the absence of lymphatic drainage from the eye, metastases form via hematogenous spread and are rarely found at the time of initial diagnosis. Due to the lack of effective treatments and the high tumor burden at the time of detection, the 1-year survival rate of UM patients drops dramatically to 15% once it has metastasized. The liver is the most common site of metastasis (60–89%), followed by lungs (24–29%), skin and soft tissue (11–12%), bone (8–17%), and lymph nodes (11%) [12,13].

The prognosis of UM has been associated with tumor location, age, and sex: indeed, iris melanoma has a 5–10-fold lower mortality rate than posterior UM [14], while younger age and female sex appear to be protective against metastatic disease [15,16]. Other relevant prognostic factors are (i) the potential infiltration of ciliary bodies, representing a marker of poor prognosis with respect to the solitary choroidal involvement; (ii) presence/absence of extraocular extension with emphasis on sclera invasion; (iii) pathologic tumor staging based on the greatest thickness and largest basal diameter of the neoplasm [17], according to the 8th edition of TNM staging. It is recommended that pathologists also report mitotic activity (number of mitoses/mm^2^) and Ki-67 (MKI67, marker of proliferation Ki-67) score as a measure of proliferative activity [18].

Despite a common melanocyte origin, UM and cutaneous melanoma (CM) are distinct diseases in terms of both genetic alterations and biological behavior [19]. Although UV exposure is clearly a major risk factor for CM [20], the contribution of UV exposure to UM pathogenesis is not well established [21]. Several studies showed a weak positive association between UV exposure and the development of UM [22,23,24], but others did not confirm this evidence [25,26,27,28]. On the contrary, it has been reported that increased UV exposure may have a protective effect, as people who work outdoors have been shown to have a lower risk of developing UM compared to those who work indoors [21]. Nonetheless, low levels of melanin in eyes and skin have been associated with UM [24,25], implying that UV radiation may play a role in UM, although considerably weaker than in CM.

Ophthalmologists use clinical examination to diagnose UM; several ancillary tests can be performed, including fundus oculi biomicroscopy, color fundus photography, ultrasonography (USG), ultrasound biomicroscopy (UBM) in case of anterior location of the tumor (iris or ciliary body), optical coherence tomography (OCT), and indocyanine green angiography (ICGA) [29]. Patients may be asymptomatic or exhibit symptoms such as blurred vision, photopsia, floaters, and loss of visual field [30]. Because UM is often asymptomatic, it is frequently detected during a routine ophthalmology examination by instrumental methods, the results of which are sometimes unclear due to an overlap in size between small UM and benign choroidal nevi. UM is one of the few cancers in which biopsy is not usually used to confirm the diagnosis. Furthermore, biopsy analysis in UM is rarely feasible, especially for diagnostic purposes, because the presumed tumor tissue must be extracted from the eye after enucleation, nullifying the diagnostic significance of the analysis. Otherwise, an intraocular tumor biopsy could be performed. The latter technique is highly debated due to the theoretical risk of tumor dissemination caused by its invasiveness; additionally, the lesions’ small size and posterior location increase the risk of insufficient sampling and sight-threatening ocular complications [31]. However, in UM lesions, biopsies are commonly used for prognosis.

Despite advancements in primary tumor treatment, metastasis rates and overall survival (OS) have remained unchanged over the past decades [6,32]. OS is approximately one year after the diagnosis of metastatic disease. In fact, patients who have metastasis at the time of primary tumor diagnosis are less likely to receive aggressive primary tumor treatments. There are two types of primary UM treatment: globe-preserving treatments (e.g., radiation therapy, laser, and surgical therapy) and enucleation [4,29,33]. Enucleation surgery is typically used to treat large tumors, multifocal and diffuse iris melanoma, poor visual function, and recurrent tumors [3]. Local tumor recurrence is typically treated in the same way that the primary tumor was [34]. Secondary orbital involvement is difficult to treat and requires radiotherapy and surgery (excision, debulking, or exenteration) [35]. Systemic monitoring is used to early detect metastasis and may have clinical implications because some selected cases of hepatic metastasis may be managed with surgical resection, resulting in improved survival [36]. Because UM has a proclivity to metastasize to the liver, surveillance imaging is often centered on hepatic monitoring [37].

The only effective therapeutic approach against metastatic disease is tebentafusp, which was approved by the FDA in January 2022 [38]. Several chemotherapeutic drugs, including cisplatin, dacarbazine, fotemustine, temozolomide, treosulfan, have been studied, with underwhelming results [11,39]. Unlike CM, immunotherapy did not improve the outcome of patients with UM [40]. One reason for such a disparity in immunotherapy response could be due to the biological and immunogenic differences between CM and UM [41,42]. Given the prevalence of *GNAQ* (G protein subunit alpha q)/*GNA11* (G protein subunit alpha 11) mutations in UM, agents targeting downstream effectors of *GNAQ*/*GNA11* pathways, such as *MEK* (now named *MAP2K7*, mitogen-activated protein kinase kinase 7) (e.g., selumetinib and trametinib) and *PKC* (now named *PRRT2*: proline-rich transmembrane protein 2) (e.g., sotrastaurin), have been studied [2]. However, as with other therapeutic approaches, unsatisfying results have been reported, with overall response rates <15% [11,39].

Recently, there has been an increase in interest in the molecular mechanisms involved in UM carcinogenesis, progression, and dissemination, which could lead to the discovery of valuable diagnostic and prognostic biomarkers, as well as new potential therapeutic targets. Cellular, genetic, and RNA-based epigenetic features of UM will be described in this review to provide a useful tool for both clinicians and researchers to better understand the molecular bases of UM.

### Histopathological Features of UM

UM frequently appears on gross examination as a dome-shaped or ring-shaped mass protruding into the posterior chamber of the eye [18,43,44]. Hemorrhagic and/or necrotic foci, as well as the presence of extraocular extension and retinal detachment, may be found in some cases [43,44]. UM can exhibit different degrees of pigmentation, ranging from highly pigmented to grayish in color mass [18]. Callender proposed the first histopathologic classification of UM based on the predominant cell component [45]. Six UM subtypes have been identified: (i) spindle A, (ii) spindle B, (iii) fascicular, (iv) mixed, (v) epithelioid, and (vi) necrotic. Spindle-shaped cells with elongated and slender nuclei, fine chromatin, small/inconspicuous basophilic nucleoli, and nuclear folding were typical of the spindle A subtype. Spindle B UMs exhibited spindled and cohesive cells with plump fusiform/cigar-like nuclei, coarse chromatin, and prominent basophilic or eosinophilic nucleoli. Fascicular UM subtype showed spindle-shaped cells arranged in a fascicular growth pattern with nuclear palisading and an overall morphology closely reminiscent of that of soft tissue schwannoma. The presence of a mixture of spindled and epithelioid cells (large, polygonal-shaped cells with abundant cytoplasm, distinct cell membranes, and large nuclei) and the absence of the spindle cell component were histologic features of the mixed and epithelioid UM subtypes, respectively. Finally, necrotic UMs were described as tumors in which extensive necrosis prevented a more accurate classification. According to Callender, the outcome of patients affected by mixed, epithelioid, and necrotic UM was poorer than that of patients with spindle A, spindle B, or fascicular tumors [45].

Because this “old” classification lacked reproducibility among pathologists and prognostic relevance, it was simplified over time, and the American Joint Committee on Cancer (AJCC) now recognizes three histologic subtypes of UM [46,47]: (i) spindle cell type (typically composed of both spindle A and spindle B cells and exhibiting spindle cell morphology in 90% of tumor); (ii) epithelioid cell type (exhibiting epithelioid cell morphology in 90% of tumor); and (iii) mixed cell type. Further unusual morphological variants of UM have been described: (a) diffuse UM, defined as tumors involving at least one-quarter of the uvea [48]; (b) clear cell UM, characterized by diffuse clear cell morphology caused by glycogen dissolution after fixation [49]; and (c) balloon cell UM, showing large tumor cells with abundant lipid-rich cytoplasm [50,51]. UM cells are commonly stained with melanocytic differentiation markers such as S-100, human melanoma black 45 (HMB45), melan-A/melanoma antigen recognized by T cells 1 (*MART-1*), melanocyte-inducing transcription factor (*MITF*) and sex-determining region Y-box 10 (*SOX10*) [52,53,54,55]. Unlike in CM, the immunoreactivity for S100 is often weak and focal in UM, and the other abovementioned markers should be preferred, accordingly [56].

Other UM histopathologic features, such as tumor-infiltrating lymphocytes (TILs) and vascular patterns, have been classically correlated with the prognosis of UM. Although TILs are less frequently found in UM than in CM, two patterns of lymphocytic infiltration have been described: patchy or diffuse [57,58,59,60,61]. High levels of TILs have been correlated with poor prognosis and chromosome 3 loss in UM patients, while it has been widely accepted that they are predictive of a better outcome in other malignancies such as breast carcinoma, CM, and non-small cell lung cancer. In addition, nine vascular patterns were described by some clinicians [62]: (i) unaffected choroid vessels; (ii) absence of tumor vessels; (iii) straight vessels; (iv) parallel vessels; (v) parallel vessels with cross-linking; (vi) arcs or incomplete loops; (vii) branching arcs; (viii) complete loops; (ix) networks of ≥3 closely packed vascular loops. Based on these findings, the admixture of epithelioid cell morphology, mitoses, and at least 1 closed loop of vessels has been reported as a relevant predictive factor of poor prognosis of enucleated UM.

## 2. Driver Gene Mutations

Chromosome and gene alterations are genetic factors that contribute to cancer’s emergence, progression, and tumor metastasis. In UM, a variety of chromosome and gene functional and numerical derangements in critical molecular pathways (such as cell cycle regulation, signaling transduction, apoptosis, or angiogenesis) have been identified and described. Particular genetic signatures at the chromosomal or gene mutation level influence tumor biology and lead to aggressive phenotypes (metastases, poor response, and low survival rates). The identification of driver mutations for diagnostic, prognostic, and therapeutic purposes has become a focal point of cancer precision medicine [63]. *BAP1* (BRCA1-associated protein 1), *EIF1AX* (eukaryotic translation initiation factor 1A X-linked), *GNA11*, *GNAQ*, and *SF3B1* (splicing factor 3b subunit 1) are the most frequently mutated genes that are thought to be drivers in UM development and progression [64,65,66,67]. To follow, these mutations and their significance in the biology of UM will be described (Table 1).


*BAP1*


*BAP1* is a tumor suppressor gene that encodes a nuclear deubiquitinase involved in cell growth and cancer pathogenesis, mapping on chromosome 3 (3p21.1) [75,76]. *BAP1* loss of function mutations or loss of expression are linked to an increased risk of metastatic disease [73]. For a long time, chromosome 3 loss was the strongest indicator of metastatic disease in UM patients. Subsequently, the identification of various gene expression profiles (GEPs) enhanced prognostic accuracy: using next-generation sequencing, it was discovered that the large majority of class 2 tumors carried a mutation in the *BAP1* gene [73]. Uner et al. suggested that *BAP1* mutations occur early in the growth of UM, well before the primary tumor is diagnosed, and coincide with the seeding of micrometastases [77].

Furthermore, it was observed that tumors with monosomy 3 and *BAP1* mutations have lower disease-free survival (DFS) rates [73]. *BAP1* encodes a nuclear ubiquitin carboxy-terminal hydrolase (*UCH*), which is one of the several deubiquitinating enzyme classes [78]. *BAP1* contains a UCH37-like domain (ULD) [79], binding domains for *BRCA1* (BRCA1 DNA repair associated) and *BARD1* (BRCA1-associated RING domain 1), which form a tumor suppressor heterodimeric complex [80], and a binding domain for host cell factor-1 (*HCFC1*), which interacts with histone-modifying complexes during cell division [73,79] by removing ubiquitin molecules from histone H2A [81]. *BAP1* also interacts with *ASXL1* (ASXL transcriptional regulator 1) to form the Polycomb group repressive deubiquitinase complex (PR-DUB), which is implicated in stem cell pluripotency and other developmental processes [73]; this activity modulates Hox (homeobox) gene expression, implying that *BAP1* regulates transcription during development. *BAP1* has also been found to be involved in other important cellular functions, such as cell proliferation via interaction (by deubiquitination) with *HCFC1*, which acts as a transcriptional coactivator with E2F proteins during cell division [82]. *BAP1* mutations were previously discovered in a small number of breast and lung cancer cell lines [78], as well as in malignant pleural mesotheliomas [83], CM [84], and possibly other cancers such as meningioma [85]. *BAP1* was discovered during a protein interaction screening for *BRCA1* and has been shown to collaborate with *BRCA1* in tumor suppression in cultured cells [78,80]. Depletion of *BAP1* in UM in vitro models leads to loss of differentiation and gain of stem-like properties, such as stem cell marker expression and an increased ability of self-replication, suggesting a role as a regulator of differentiation of uveal melanocytes [86]. No correlation between *GNAQ* and *BAP1* has been reported in literature [73,76]. In rare cases, a germline *BAP1* mutation can be found. These mutations are associated not only with an increased risk of UM but also with other types of tumors [87].

*GNAQ* and *GNA11*

*GNAQ* and *GNA11* genes map on chromosomes 9q21.2 and 19p13.3, respectively. They are paralogous genes with roughly 90% sequence homology and a coding region of seven exons [68].

*GNAQ* encodes the alpha q subunit (Gαq), while *GNA11* encodes the alpha 11 subunit (Gα11), both of which are guanine nucleotide-binding proteins from the heterotrimeric protein family that are important in transmembrane signaling networks. The alpha subunits act as a switch between the active and inactive states of G-proteins, active when bound to guanosine triphosphate (GTP) and the inactive when GTP is hydrolyzed to guanosine diphosphate (GDP) [88]. *GNAQ* and *GNA11* genes regulate several intracellular pathways, including the rapidly accelerated fibrosarcoma (*RAF*)/mitogen-activated protein kinase kinase (*MEK*)/extracellular signal-regulated kinase (*ERK*) pathway. When the *RAF*/*MEK*/*ERK* pathway is activated, the cell-cycle regulatory protein cyclin D1 (*CCND1*) is overexpressed, which leads to the inactivation of the tumor suppressor *RB1* (RB transcriptional corepressor 1) [67].

*GNAQ*/*GNA11* mutations are present in approximately 80–90% of UM cases [89,90]. Alterations in the downstream pathway are thought to be an early event in the development of cancer, as they activate multiple cascade pathways involved in cell growth and proliferation [91]. *GNAQ* mutations have been found in all stages of malignant progression, indicating that it is an early event in UM, but it is unrelated to DFS [90]. The oncogenic conversion of *GNAQ* results in constitutive activation of the MAPK (mitogen-activated protein kinase) pathway, culminating in a situation in which the cell receives continuous growth signals in the absence of extracellular stimuli, resulting in cell proliferation [92,93]. *GNA11* mutations, on the other hand, can occur at various stages of UM progression. The majority of uveal nevi has either *GNAQ* or *GNA11* mutations, with *GNA11*-mutated tumors being potentially more aggressive than *GNAQ*-mutated ones. This is most likely due to the fact that, unlike *GNA11*, *GNAQ* requires a second hit to fully activate [94]. The first driver mutations in UM are thought to be *GNAQ*/*GNA11*-activating mutations. Mutations in *GNAQ* and *GNA11* occur in a mutually exclusive pattern and are found almost exclusively in codon 209 and, in some cases, codon 183. Mutations at these positions result in constitutive activation of the Gαq and Gα11 subunits by removing their intrinsic GTPase activity, preventing the subunits from reverting to an inactive state. When constitutively activated, both *GNAQ* and *GNA11* have been shown to upregulate the MAPK pathway in the same way that *BRAF* (B-Raf proto-oncogene, serine/threonine kinase) and *NRAS* (NRAS proto-oncogene, GTPase) mutations do. Activating mutations in *BRAF* are very common in CM, whereas UM rarely carries any *BRAF* mutation. [95,96]. Mutations in *GNAQ* and *GNA11* have not been linked to the two GEP molecular classes of UM tumors. Furthermore, *GNAQ*/*GNA11* mutations have not been shown to be prognostic, and they occur at similar rates in metastatic and non-metastatic tumors. Mutational hotspots in both genes have already been reported in the literature, defined by the presence of activating missense variants that only affect exons 4 and 5, and more precisely, the arginine 183 (R183) and glutamine 209 (Q209) codons; cell lines with *GNAQ* Q209L mutation have also been found to be highly sensitive to *MEK* inhibition [66]. The huge percentage of *GNA11* codon 209 mutations results in glutamine to leucine (p.Q209L) and proline (p.Q209P) substitutions [68,89,97,98]. These mutations result from one-base substitutions at codon 209 (CAG), with A>T (94.5%) and A>C (2.7%) being the most prevalent [65]. In contrast, a one-base change at codon 209 (CAA) in *GNAQ* gene leads to the substitution of glutamine by leucine (A>T, p.Q209L) and proline (A>C, p.Q209P) in the majority of cases [66,68]. Other mutations in exon 5 have been described, including p.Q209M, p.Q209H, p.Q209I, p.F228L, and p.M203V in *GNAQ* and p.Q209Y, p.E234K, and p.E221D in *GNA11* [68,99]. Overall, the frequency of mutations in *GNAQ* and *GNA11* exon 4 is lower. Most mutations in *GNA11* are induced by C>T transitions at codon 183 (CGC) and CC>TT transitions at codons 182–183, that cause arginine to cysteine (p.R183C) or histidine (p.R183H) replacement. Similarly, the few mutations known to affect *GNAQ* codon 183 (CGA) are all induced by G>A transitions [66]. Other *GNAQ* exon 4 mutations are p.P170S, p.I189T, p.Q176R, and p.P193L, which have an overall recurrence of 8.9% in some cohorts [99]. Mutations in the codon p.Q209 result in the complete loss of GTPase activity, causing a prolonged constitutive activation of *GNAQ* and *GNA11*, which leads to permanent downstream signaling. Mutations that affect the p.R183 residue, on the other hand, result in a more tenuous activation due to a partial loss of GTPase activity [66].

Less is known about the prevalence and significance of *GNAQ* and *GNA11* mutations in metastatic UM (MUM) [68]. The first studies on the role of these genes in UM prognosis reported that the distribution of *GNA11* and *GNAQ* mutations varied between primary tumors and MUM, with a *GNA11* to *GNAQ* ratio of 0.7 in primary UM and 2.6 in MUM [65,68]. Griewank et al. discovered that *GNA11* mutations were significantly more common than *GNAQ* mutations in metastatic specimens [100]. Furthermore, patients with *GNA11*-mutant tumors had lower disease-specific survival and OS compared to wild-type patients. The authors proposed that the survival data, combined with the predominance of *GNA11* mutations in metastasis, raises the possibility that *GNA11*-mutant tumors may be associated with a higher risk of metastasis and a worse prognosis than *GNAQ*-mutant tumors [68,100].

Terai et al. recently investigated the existence of a relationship between metastasis-to-death and the frequency of *GNAQ* and *GNA11* mutations in eighty-seven MUM patients. The authors reported a similar mutation rate for *GNA11* and *GNAQ* mutations (47.1% and 44.8% of patients, respectively) [101]. This result was consistent with previous findings for primary UM [65,66]. Moreover, they discovered that differences in the type of mutation (p.Q209 vs. p.Q209L) rather than the *GNAQ* and *GNA11* genes themselves could predict MUM patient survival [68,101].

Functional differences between *GNA11* and *GNAQ* might be determined by different interaction partners. To investigate this aspect, Piaggio et al. conducted a study using tandem affinity purification and mass spectrometry (TAP-MS/MS) [102] to identify proteins that interact with *GNAQ* or *GNA11*. The comparison of the protein interaction networks of the two Gα-proteins only showed a very limited overlap, indicating functional differences between *GNAQ* and *GNA11*. The interaction of mutated *GNAQ* with the dioxygenase *TET2* (tet methylcytosine dioxygenase 2), which is not observed for mutated *GNA11*, was confirmed by coimmunoprecipitation analyses. Interestingly, *TET2* plays an active role in DNA demethylation, and high-risk UMs are characterized by widespread demethylation [102].

Another alteration caused by *GNA11* and *GNAQ* mutations concerns the calcium signaling pathway, whose dysregulation has a well-documented association with cancer survival, proliferation, migration, and metastatic potential. For example, calcium signaling has been reported to be involved in the proliferation of Ras-driven cancers through the interaction between calmodulin and *PI3K* (phosphatidylinositol-4,5-bisphosphate 3-kinase) [103,104] and the promotion of invasion and metastasis via *ERK* activation in both *BRAF*- and non-*BRAF*-driven CM cells [104,105]. Constitutive activation of Gαq signaling by mutations in *GNAQ* or *GNA11* occurs in over 80% of UMs and activates MAPK signaling [106]. Chen et al. reported that Ras oncoproteins are required for *GNAQ*-mediated MAPK activation and identified *PRKCD* (protein kinase C delta), *PRKCE* (protein kinase C epsilon), and *RASGRP3* (RAS guanyl-releasing protein 3) as components of a signaling module necessary and sufficient to activate the Ras/MAPK pathway in *GNAQ*-mutant UM [106]. *RASGRP3* is selectively overexpressed in response to *GNAQ*/*GNA11* mutations in UM; its activation occurs via *PRKCD*- and *PRKCE*-dependent phosphorylation and *PKC*-independent, DAG (diacylglycerol)-mediated membrane recruitment, possibly explaining the limited effect of *PKC* inhibitors in durably suppressing MAPK in UM. The results achieved by Chen et al. suggested *RASGRP3* as a therapeutic target for cancers driven by oncogenic *GNAQ*/*GNA11* [106].


*EIF1AX*


*EIF1AX*, mapping on the chromosome Xp22, encodes for the X-linked eukaryotic translation initiation factor 1A (*EIF1A*) protein, which regulates protein translation initiation through a combination of ribosome stabilization and recognition of target mRNA, thereby preparing mRNA for translation [71]. *EIF1AX* is required for the transfer of methionyl initiator tRNA to the small ribosomal unit (40S) during the initiation phase of translation in eukaryotic cells [107]. Whole-exome sequencing identified *EIF1AX* as an UM driver gene [64]. This gene is mutated in approximately 14–20% of all UM, with the majority of mutations found in exons 1 and 2 [64,108]. *EIF1AX* mutations are typically found in non-metastatic cases, are associated with class 1 GEP tumors and a favorable prognosis, and are inversely correlated with metastasis [109,110]. *EIF1AX* mutations are typically reciprocally exclusive with *BAP1* mutations and, to a lesser extent, with *SF3B1* mutations. Most *EIF1AX* mutations are observed in tumors with disomy 3 (48%), and only rarely in tumors with monosomy 3 (3%) [64]. In contrast to *BAP1* mutations, which mainly are truncating and loss-of-function variants, the majority of *EIF1AX* mutations are heterozygous non-synonymous variants, or in some cases splicing variants, leading to deletions of one or two amino acids; thus, in most cases, the core protein remains unchanged [71]. *EIF1AX* mutations are commonly found in tumor cells in heterozygosis, indicating that *EIF1AX* functions as a dominant-acting oncogene. However, UM tumors with an *EIF1AX* mutation only express the mutant allele, suggesting that *EIF1AX* may also function in a recessive manner [64]. Mutations in this gene have also been found in thyroid and ovarian cancers, as well as the rare neoplasm primary leptomeningeal melanocytic neoplasms (LMNs) [111,112].


*SF3B1*


*SF3B1* encodes a core component of the RNA splicing machinery, the spliceosome, which processes precursor mRNA into mature transcripts, and maps on chromosome 2q33 [113]. *SF3B1* mutations are mostly identified in hematolymphoid malignancies. Mutations in codon 700 represent 50% of all of the reported alterations; other mutations were found in codons 666, 662, 622, and 625 [114]. In UM, *SF3B1* mutations almost exclusively occur in codon 625 and have been identified in 4% to 24% of primary tumors [110,115]. Accordingly, it is yet another driver gene discovered through whole-exome sequencing of UM tumors. *SF3B1* is required for pre-mRNA splicing because it encodes a unit of the splicing factor 3b protein complex, which is a component of both major (U2-like) and minor (U12-like) spliceosomes [113]. *SF3B1* has been identified in recent years as a DNA damage repair factor [116]. Missense mutations in specific regions of the *SF3B1* gene have also been reported to alter the splicing of many target genes [71]: these mutations principally modify codon Arg625 in exon 14 and have been observed in UM tumors with mutation rates ranging from 10% to 21% [64,117]. By using RNA-seq analyses of UM, Alsafadi et al. showed that the *SF3B1* mutations resulted in deregulated splicing at a subset of splice junctions, mostly by the use of alternative 3′ acceptor splice sites (3′ss) [118]. At first, they observed that *SF3B1* hotspot mutations in UM were associated with the deregulation of a restricted subset (~0.5%) of splice junctions, mostly caused by the usage of alternative 3′ss (AG′) upstream of the canonical 3′ss (AG). Second, they showed that splicing alterations induced by *SF3B1* mutations were not reproduced either by knockdown or by overexpressing the wild-type protein, indicating that *SF3B1*-mutants may be qualified as change-of-function mutants. Third, their results provided significant progress in understanding the molecular mechanisms underlying alternative 3′ss regulation by mutated *SF3B1* [118]. Such a mechanism involves a misregulation of branchpoint (BP) usage, which has been largely overlooked in previous studies on alternative splicing [118,119].

Some studies related *SF3B1* mutations to a better prognosis, a younger age at diagnosis, and the presence of disomy 3 [117]. Moreover, in a longer-term study, tumors with disomy 3 and an *SF3B1* mutation had a significantly worse prognosis and more frequent development of late metastasis than wild-type tumors; most metastases occurred later than 5 years after diagnosis [110]. According to Martin et al., 29% of disomy 3 tumors carried a heterozygous *SF3B1* mutation, compared with only 3% of monosomy 3 tumors. In addition, 54% of partial monosomy 3 tumors (preferentially with 3q loss and preservation of 3p) carried the Arg625 mutation in *SF3B1* [64]. *SF3B1* mutations are frequently found in tumors expressing the *PRAME* (PRAME nuclear receptor transcriptional regulator) oncogene. *PRAME* expression has been linked to class 1 tumors with an intermediate risk of metastasis, implying that there is a risk class that occurs between high-risk tumors characterized by *BAP1* mutations and low-risk tumors frequently harboring *EIF1AX* mutations [120].

Other mutations

A recurrent gain-of-function mutation in the phospholipase C, beta 4 (*PLCB4*) gene was discovered using whole-genome and whole-exome sequencing of UM tumors [108]. In addition to the known driver genes in UM, this was the only gene with a recurrent mutation. The mutation (c.G1888T, p.D630Y) is located in the Y-domain of *PLCB4*’s highly conserved catalytic core and was predicted to be harmful using the prediction tools SIFT and PolyPhen [71]. *PLCB4* is a downstream target of *GNA11*/*GNAQ*, and the p.D630Y *PLCB4* mutation was found to be mutually exclusive with mutations in *GNA11* and *GNAQ* [71,108].

*CYSLTR2* (cysteinyl leukotriene receptor 2) encodes for the G protein-coupled receptor cysteinyl-leukotriene receptor 2. Different authors found a recurrent gain of function mutation in *CYSLTR2* (p. L129Q) leading to a *CYSLT2R* mutant protein constitutively activating endogenous Gαq, which is a signaling pathway convergent with the one induced by *GNAQ* and *GNA11* oncogenic mutations [121,122]. Mutations of *CYSLT2R* show a pattern of mutually exclusive activating mutations (*GNAQ* and *GNA11*) in almost all tumors and appear to be an early oncogenic event in *GNAQ* and *GNA11* wild-type uveal nevi and UM [122,123]. Mutant *CYSLT2R* increased the expression of the melanocyte-lineage-specific transcriptional program and promoted tumorigenesis in vitro and in vivo [121,122,124].

*SRSF2* (serine/arginine-rich splicing factor 2) encodes serine/arginine-rich proteins that bind exonic splicing enhancers. *SRSF2* mutations result in misregulated exon inclusions that cause an aberrant splicing pattern of many genes, including the tumor suppressor genes *ARMC10* (armadillo repeat containing 10) and *EZH2* (enhancer of zeste 2 polycomb repressive complex 2 subunit). *SRSF2* mutations are also commonly found in chronic myelomonocytic leukemia (47%) and myelodysplastic syndrome (15%) [125]. In their study, van Poppelen et al. detected *SRSF2* deletions affecting amino acids 92–100 in two UMs (5%) of 42 selected tumors and in three The Cancer Genome Atlas (TGCA) UM specimens [126]. Both the samples with an *SRSF2* mutation from their cohort and the ones from the TCGA showed more than four structural chromosomal aberrations, including a partial gain of chromosomes 6 and 8, although monosomy 3 was observed in two TCGA UMs [126].

Telomerase reverse transcriptase (*TERT*) is a component of the telomerase enzyme that adds the telomere repeat TTAGGG to the telomere ends. Telomerase deregulation and abnormal *TERT* expression have been found in a variety of cancers, including thyroid and bladder cancers [127]. Approximately 70% of CM have mutations that impact *TERT* expression levels by creating a new binding site for the transcription factor E-twenty-six (*ETS*) [128]. Following the discovery of driver mutations in the promoter of the *TERT* gene associated with UV-induced cytidine-to-thymidine transitions in CM, Dono et al. investigated the presence of this event in UM, discovering that *TERT* promoter mutations are extremely rare in UM tumors: indeed, only one out of the fifty patients in their study had one of the previously described *TERT* promoter mutations [74]. The promoter mutation was found in this study along with mutations in *GNA11* and *EIF1AX*, as well as two chromosome 3 normal copies.

*MBD4* (methyl-CpG binding domain 4, DNA glycosylase), mapping on chromosome 3, encodes for a DNA glycosylase involved in the repair of C>T mutations arising from spontaneous deamination of 5-methylcytosine. Given the high frequency of chromosome 3 monosomy in UM (see below Section 3), *MBD4* is often present in a single copy; in this scenario, a single mutation is sufficient for the inactivation of *MBD4*. Hence, *MBD4* may act as a tumor suppressor in UM [129,130]. Accordingly, Derrien et al. reported that some UMs display a high level of CpG>TpG mutations in association with the mutational inactivation of *MBD4* [129]. In particular, germline protein truncating variants (PTVs) and somatic loss of the wild-type allele were reported in UM patients with a CpG>TpG mutator phenotype. *MBD4* was suggested as a new predisposing gene for UM; indeed, it was associated with hypermutated tumors with monosomy 3 and conferred a predisposition to high-risk tumors [129]. Recently, *MBD4* was reported as a prognostic factor for response to immune checkpoint inhibitors in MUM patients [131].

Furthermore, UM has been linked to a variety of cancer-related genes in solid tumors, including *BRCA1* (c.C2603G: pSer868), *CHEK2* (c.T470C:p.Ile157Thr), *PALB2* (c.49–1G>A), *SMARCE1* (c.373G>T: p.Glu125*), *MSH6* (c.C2731T: p.Arg911), and *MLH1* (c.200G>A:p.Gly67Glu) [132].

## 3. Chromosomal Aberrations

In comparison to many other cancer types, most UMs have relatively low levels of genomic instability and aneuploidy. Recurrent chromosome aberrations in chromosomes 1, 3, 6, 8, 9, and 16 characterize primary UM. These cytogenetic changes are strongly linked to prognosis and are used to categorize patients into risk groups. 1p loss (28–34%), 1q gain (24%), 3 loss (50–61%), 6p gain (28–54%), 6q gain (28–54%), 6q loss (25–38%), 8p loss (17–28%), 8q gain (36–63%), 9p loss (24%), and 16q loss (16%) are the most common chromosomal aberrations in primary UM [133,134,135]. These abnormalities were initially identified by standard karyotypic analyses [136,137] and then confirmed by fluorescence in situ hybridization (FISH) [138], comparative genomic hybridization (CGH) [134,139,140,141], spectral karyotyping [142], microsatellite analysis (MSA) [143], multiplex ligation-dependent probe amplification (MLPA) [144], and single-nucleotide polymorphism (SNP) analysis [145].

Chromosome 1

A quarter of UMs have a partial or complete loss of chromosome 1p, which is more common in concert with monosomy 3 [146]. Kilic et al. demonstrated that loss of the short arm of chromosome 1 (1p36) in conjunction with monosomy 3 is prognostic: indeed, when these abnormalities occur concurrently, they have a stronger correlation with reduced survival than monosomy 3 or loss of 1p alone, the latter having no prognostic value [147]. *APITD1* (now named *CENPS*, centromere protein S), one of the suggested tumor suppressor genes mapping on the 1p36 region, was found to have no effect on patient survival [148]. The smallest common region of 1p loss was identified as a region of about 55 Mb at 1p31 by microsatellite analysis of seventy UMs [146]. No mutations were found in this region, but there are several potential candidates, including Notch signaling pathway members *HES2* (hes family bHLH transcription factor 2) and *HES5* (hes family bHLH transcription factor 5), as well as the *TP53* (tumor protein p53) homolog *TP73* (tumor protein p73) [147].

Chromosome 3

The most common chromosomal aberration in UM is the loss of one of the two copies of chromosome 3. Monosomy 3 is found in approximately 50% of cases [143,149] and appears to be very specific to UM, as it is rarely found in other cancer types [135]. For nearly twenty-five years, several groups have demonstrated a strong correlation between monosomy 3 and metastasis development [150]; indeed, metastases rarely develop in tumors with disomy 3 [3,151]. Furthermore, monosomy 3 is strongly associated with a number of clinical and histopathological parameters, including epithelioid cytology, closed vascular patterns, large tumor diameter, and ciliary body involvement [139,143,150]. Furthermore, monosomy 3 is thought to be an early event in tumorigenesis because it frequently occurs in conjunction with all other known chromosomal abnormalities [136]. In 5–10% of cases, one copy of chromosome 3 is lost and the remaining copy is duplicated. This chromosome 3 isodisomic state appears to be prognostically equivalent to monosomy 3 [145]. Clinical outcomes do not differ significantly between patients with partial monosomy 3 or disomy 3 [152]. When chromosome 3 has a normal copy number, tumors can have other chromosomal alterations, such as 6p gain and 1p loss [134,151]. It is important to underline that the *BAP1* locus is located at position 3p21.1; the importance of *BAP1* in the progression of UM has been discussed above (Section 2).

Chromosome 6

Chromosome 6 alterations are common in UM; gain of 6p and loss of 6q occur in about a quarter to a third of UMs; both abnormalities are frequently present in the same tumor, implying the formation of an isochromosome 6p [139]. The first chromosomal aberration to be reported in UM was gain of the short arm of chromosome 6 (6p), which has a lower prognostic value than monosomy 3 or gain of 8q [135,153]. Gain of the short arm of chromosome 6 is found in 28–54% of UMs and is associated with spindle cell cytology and a low risk of metastasis [135,136,154,155,156]. However, the simultaneous occurrence of monosomy 3 or 6p gain is rare: they are most likely involved in two mutually exclusive evolutionary pathways, as the occurrence of both is reported in only 4% of UMs [140,141,157,158]. Overall, 6p gain has a better prognosis than monosomy 3, leading some researchers to speculate that 6p gain is “protective” against metastasis [159]. However, 6p gain appears to be associated with a better prognosis simply because it occurs in the absence of monosomy 3 [140]. Loss of genetic material on the long arm of chromosome 6, observed in 25–38% of tumors, possibly represents another late event in tumorigenesis and correlates with poor prognosis [135,139,156,160,161].

Chromosome 8

Chromosome 8 is also frequently altered in UM patients [162]. Gain of the long arm of chromosome 8 (8q) occurs in 37% to 63% of primary UM [134,163,164,165,166] and is associated with poor prognosis. Gain of 8q has been shown to be a significant independent prognostic factor for shorter survival [167,168]. It is frequently found in conjunction with monosomy 3, either as an 8q gain or as an 8q isodisomy, and this combination is associated with higher metastatic rates than a single aberration [141,169]. Chromosome 3 and 8 abnormalities are more common in ciliary body-located UMs, whereas alterations of the long arm of chromosome 8 are more common in choroid-derived UMs [136,139,153,167]. However, in the study by Kilic et al., chromosome 8q abnormalities were shown to correlate with large tumor diameter, but univariate analysis revealed no significant relationship between 8q gain and the metastatic phenotype, suggesting that it is a late event after the onset of monosomy 3 [134]. The 5-year mortality rate is reported to be 66% in cases of concomitant monosomy 3 and 8q gain, 40% in cases of monosomy 3, and 31% in cases with 8q gain [2,143]. Dogrusoz et al. investigated whether chromosome 3 and 8q status information could improve the prognostic value of the AJCC staging system: tumors with monosomy 3 and 8q gain had an increased risk of metastatic death in the study cohort of 470 UMs with known chromosome 3 and 8q status [165].

It has not yet been determined which chromosomal change causes UM malignant transformation. Some researchers discovered that monosomy 3 is the first step and that 8q gain occurs later [140], while others discovered that 8q gain occurs before chromosome 3 loss [170]. Finally, other researchers reported that the gain of the telomeric part of 8q is present in 92% of the UM studied, implying that it plays an important role in UM tumorigenesis [171]. Several oncogenes on chromosome 8q have been suggested as potential factors involved in UM, among which *MYC* (MYC proto-oncogene, bHLH transcription factor) (on 8q24), *NBS1* (now named *NBN*, nibrin) (on 8q21), and *DDEF1* (now named *ASAP1*, ArfGAP with SH3 domain, ankyrin repeat, and PH domain 1) (on 8q24) [153,172,173,174,175]. Furthermore, in 50% of UMs, the *NBS1*/*NBN* gene is found to be overexpressed [173]: the encoded protein is thought to be a component of a complex involved in DNA repair [176]. Overexpression of *NBS1*/*NBN* may allow UM progression by promoting DNA repair, which happens more frequently in advanced-stage tumors with increased genetic instability. Ehlers et al. demonstrated that high *DDEF1*/*ASAP1* (expression results in more motile low-grade UM cells and may thus be important in metastatic development [174].

Onken et al. identified a potential metastasis-suppressor gene in *LZTS1* (leucine zipper tumor suppressor 1), mapping on 8p21; they also discovered that 8p loss was a better prognostic factor than 8q gain [177]. Thus, 8p loss may be more significant than 8q gain, both prognostically and pathogenetically.

Chromosome 9

Almost a quarter of UMs have a cytogenetically detectable loss of chromosome 9p, and smaller regions of loss of heterozygosity (LOH) around 9p21, including the *CDKN2A* (cyclin-dependent kinase inhibitor 2A) locus, are found in up to a third of UMs [161]. The *CDKN2A* promoter is methylated in 24–31% of cases [178,179]. These findings suggest that *CDKN2A* inactivation may play a role in the progression of UM. However, germline *CDKN2A* mutations are extremely rare in UM patients [180,181].

Other Chromosomal Aberrations

Other less common aberrations are chromosome 16 abnormalities, specifically loss of 16q arm (16% of cases) [134]. Loss of chromosome 10, loss of 11q23-q25, and gain of chromosomes 7 and 10 have been reported [149,150,153,160], but a role in tumorigenesis and/or metastasis in UM is yet to be determined.

Lalonde et al. evaluated the clinical relevance of low-frequency copy number aberrations (CNAs) in UM [182]. Their study, based on the genomic profiling of 921 primary tumors, revealed CNAs associated with the risk of metastasis and demonstrated a strong association between chromosomal instability and patient prognosis. Their results suggested that 1p and 16q deletions should be incorporated into clinical assays to assess prognosis at diagnosis and guide enrollment in clinical trials for adjuvant therapies [182].

## 4. Gene Expression Profile Classification of UM

Melanomas (both CM and UM) show a high frequency of metastatization, dramatically affecting patient survival and therapeutic approaches given the acquired drug resistance [183]. Hence, it is fundamental to predict which tumors will develop metastasis to select the best clinical approach for each patient. Unfortunately, melanomas lack a definite staging because of the manifestation of heterogeneous alterations at clinical, cytologic, and morphologic levels [184,185]. For such reasons, markers of predisposition to metastatization have been investigated.

In this scenario, GEPs have been analyzed in UM biopsies. The first report was published in 2003, when Tschentscher et al. reported the differential expression of 7902 genes detected by using an oligonucleotide microarray in twenty primary UM samples, comparing tumors with monosomy or disomy of chromosome 3 [186]. Indeed, monosomy 3 is highly associated with metastatization (see above Section 3). The authors showed that the expression profile of the 7902 genes allowed to discriminate two different groups of samples, one comprising nine out of the ten disomic tumors and the other with all the monosomic plus the last disomic tumors. No significant association was observed between these clusters and chromosomal aberrations or clinicopathological features. The same clusterization was confirmed when performing the same analysis excluding the genes mapping on chromosome 3, which could have altered the analysis because of the different dosages in monosomic tissues. Only a sample showed variable classification in bootstrap sampling; interestingly, this sample was not classified as consistent concerning chromosome 3 status. Moreover, clusterization was maintained by randomly reducing the number of genes, reaching a minimum of 300 random genes. Collectively, these results suggest that GEP plays an important role in the phenotype and classification of UM. The high number of dysregulated genes suggests that the two tumor clusters are deeply different from a molecular point of view, making them two different entities. The different status of chromosome 3 may suggest a different pathogenetic process sustaining UM onset. It is also conceivable that tumors originate from melanocytes from distinct regions of the uveal tracts (iris, ciliary body, or choroid), as it is known that tumors located in the posterior chamber of the eye or in the ciliary body differ in chromosome 3 status. The limits of this study were the recruitment of patients with a recent diagnosis and the lack of follow-up [186].

In 2004, Onken et al. performed gene expression profiling in primary UM biopsies from patients followed up for a long period [154]. Microarray analysis allowed to identify 3075 genes expressed in twenty-five tumor samples. Principal component analysis of gene expression data defined two clusters comprising fourteen and eleven samples, respectively, named class 1 and class 2. Among the 3075 expressed genes, 62 were identified as discriminating between the two clusters, including some genes mapping on chromosome 3 and 8q that showed decreased and increased expression, respectively, in class 2. This observation is congruent with literature data reporting the association of metastasis with loss of chromosome 3 and gain of 8q (see above Section 3); nevertheless, a more stringent analysis led to the exclusion of most of these genes from the results. This molecular classification showed a correlation with increased patient age, a feature known to be associated with an increased risk of metastasis development. It was also observed that there was a strong positive correlation with the cytologic rank, with class 1 biopsies being low-grade spindle tumors and class 2 high-grade tumors with a higher proportion of epithelioid cells. The authors reported that multiple signatures of three genes were sufficient for a correct classification of tumors without errors, with *PHLDA1* (pleckstrin homology-like domain family A member 1), *FZD6* (frizzled class receptor 6), and *ENPP2* (ectonucleotide pyrophosphatase/phosphodiesterase 2) representing one of the best combinations of discriminating genes. The possibility of analyzing a small number of genes may pave the way for clinical applications of this classification. This molecular signature was used to classify a total of fifty tumors; subsequent survival analysis by Kaplan–Meier curves showed one and eight deaths for metastasis in classes 1 and 2, respectively, with a higher survival probability for class 1 patients. Importantly, the molecular signature was the best survival predictor when compared to the other clinicopathological prognostic factors. A strong correlation between the molecular classification and chromosome 3 and 6p aberrations was reported; noteworthy, the gene expression profile may improve the identification of high-risk patients compared to the sole analysis of chromosome status [154]. The same group analyzed the pathways regulated by the discriminating genes, showing that their altered expression between class 1 and class 2 tumors may be the reason for the different cytologic morphology observed [187]. In the following years, several studies reported that GEP outperformed chromosome 3 monosomy and other clinicopathological parameters in predicting metastasis [188,189,190,191]; an enhancement of the prognostic performance of GEP was reported when it was associated with tumor size measurement [192]. The association between GEP and chromosomal aberrations in UM was also evaluated, showing that *BAP1* and *EIF1AX* mutations were associated with class 2 and class 1 tumors, respectively [72]. However, Stålhammar and Grossniklaus reported that *BAP1* expression was heterogeneous in UM tumors and showed limited prognostic importance, despite adding significant prognostic information to GEP [193].

Aiming to apply GEP classification to clinical practice, Onken et al. identified a signature of fifteen genes (twelve discriminating and three endogenous control genes) that can be analyzed in PCR in small samples such as fine needle aspirate biopsies, characterized by low quantities of RNA [194,195]. The introduction of such a prognostic test in clinical practice would allow to identify patients with a high risk of metastasis and improve their clinical management. A prospective multicenter study [196] and an independent report from another research group [197] confirmed the efficacy of this signature. In recent years, another group identified a new molecular signature comprising ten genes by using expression data from TCGA [198] and Gene Expression Omnibus (GEO) DataSets (GSE22138 [199]); again, the signature showed increased accuracy in predicting overall, progression-free, and metastasis-free survival compared to other prognostic parameters [200].

The same approach was also applied to non-coding RNAs (ncRNAs), including microRNAs (miRNAs). Following other reports from the same group, Worley et al. performed a microarray analysis in a cohort of twenty-four UM tissues. MiRNA expression allowed to classify the tumors in low (class 1) and high (class 2) risk of metastasis, and this classification was coherent with the one based on GEP. A set of sixty-eight miRNAs showed increased expression in class 1 tumors, while six were upregulated in class 2; the best discriminator miRNAs were let-7b and miR-199a. The set of six miRNAs upregulated in class 2 tumors, including let-7b, miR-199a, miR-199a*, miR-143, miR-193b, and miR-652, showed the highest accuracy in sample classification, with 100% sensitivity and specificity [201]. Another study with a similar aim was performed in a cohort of twenty-six UM tissues classified as low-, intermediate-, and high-risk of metastasis according to DFS and mutational status of *EIF1AX*, *SF3B1*, and *BAP1*, respectively. MiRNA expression analysis by sequencing allowed to identify three sample clusters, one for each sample group, with the intermediate cluster partially overlapping the other two. The metastasis-related miRNAs showing differential expression in high- compared to low- and intermediate-risk tumors were thirteen, with overexpression of miR-132-5p, miR-151a-3p, miR-17-5p, miR-16-5p, and miR-21-5p, and downregulation of miR-181b-5p, miR-101-3p, miR-378d, miR-181a-2-3p, miR-99a-5p, let-7c-5p, miR-1537-3p, and miR-99a-3p. A similar clusterization was observed with sequencing data of mRNAs, with a separate cluster for high-risk tumors and a partial overlap of low- and intermediate-risk tumors [202]. Recently, ferroptosis, an iron-dependent mechanism of programmed cell death, has gained growing attention in cancer research. Jin et al. analyzed expression data from the TCGA and GSE84976 datasets [203] to identify ferroptosis regulators showing differential expression in patients grouped according to OS. According to the expression of eleven genes (*CHAC1*, *NQO1*, *SQLE*, *SLC1A5*, *GSS*, *LPCAT3*, *GPX4*, *AIFM2*, *ABCC1*, *ACSF2*, *FDFT1*), tumor samples were classified into two clusters differing in prognosis and tumor microenvironment-infiltrating immune cells [204]. Zheng et al. investigated the methylation of CpG islands associated with miRNAs in TCGA samples, showing fifty-five CpG sites with altered methylation in patients with different survival time. The authors identified a prognostic signature of thirteen miRNA-associated CpG sites able to classify patients into low- and high-risk groups [205]. Altered DNA methylation at ncRNA loci has been previously reported in other cancer models [206,207].

## 5. NcRNA-based Epigenetic Mechanisms in UM

NcRNAs are a heterogeneous class of RNA molecules that are not transcribed into proteins. Indeed, the function of these transcripts is to regulate several physiological and pathological processes with many different molecular mechanisms, most of which are still under investigation and may be classified as epigenetic mechanisms. NcRNAs may be classified according to length into: (i) small ncRNAs, including, among others, miRNAs, sized less than 200 nucleotides (nts) and all sharing the same molecular mechanism; (ii) long non-coding RNAs (lncRNAs), including circular RNAs (circRNAs), ranging from 200 nts to kilobases in length and characterized by heterogeneous functions and molecular mechanisms [208].

### 5.1. Small ncRNA-Mediated Epigenetics in UM

Since the completion of the Human Genome Project in 2003, ncRNAs have gained attention as strong epigenetic regulators of both physiological and pathological processes within cells. The most widely investigated class of ncRNAs is represented by miRNAs, small endogenous single-stranded RNA molecules sized 18–25 nts, which act as negative regulators of gene expression at post-transcriptional level [209]. The crucial role of miRNA function in carcinogenesis is widely recognized, as demonstrated by the huge amount of literature published in the last few decades [208,210,211,212,213,214]. Several studies reported the dysregulation of miRNAs in virtually all cancer models, proving that altered expression of these ncRNAs affects cell phenotype and cancer-related processes via epigenetic regulation of mRNA expression. In this context, UM is not an exception. Besides the most common analysis comparing tumor and normal tissues, several studies focus only on tumor tissues or start with expression data from TCGA (which includes only tumor samples in the UM dataset), aiming to investigate the prognostic value of miRNA dysregulation (Table 2).

#### 5.1.1. Prognostic miRNAs

The first study investigating altered miRNA levels was published in 2008 by Worley et al., proving that miRNA expression profile allowed to classify tumors according to the risk of metastasis. This analysis allowed to identify a signature of miRNAs with prognostic value including let-7b, miR-199a, miR-199a*, miR-143, miR-193b, and miR-652, all upregulated in high-risk tumors (see above Section 4 [201]). Venkatesan et al. analyzed UM tumors grouped according to chromosome 3 monosomy or disomy [224]. Microarray expression data obtained from six formalin-fixed, paraffin-embedded (FFPE) samples was subject to supervised and unsupervised analysis, showing fourteen upregulated miRNAs in common. Among these miRNAs, miR-149*, miR-1238, and miR-134 were validated in real-time PCR; no significant difference was observed for miR-149* and miR-1238, while miR-134 was upregulated in monosomic tumors. All these miRNAs showed increased expression in tumor samples compared to five samples of normal melanocytes from cadaveric eyes, regardless of chromosome 3 status or the presence of liver metastasis [224]. The same study also investigated the expression of miRNAs used to classify tumors by Worley et al. [201]: miR-214, miR-143, miR-146b, and miR-199a showed increased expression in monosomic compared to disomic tumors, while no difference was observed for let-7b. Moreover, miR-134 and miR-149* showed a significant association with liver metastasis. Metastasis-free survival analysis showed that all the dysregulated miRNAs may be used as prognostic factors in UM patients [224]. However, the unfeasibility of miRNAs as prognostic biomarkers was reported in a cohort of twenty-six Danish patients [258]. A similar study by Triozzi et al. showed six upregulated (miR-135a*, miR-624, miR-449b, miR-142-5p, miR-92b, miR-628-5p) and nineteen downregulated miRNAs (miR-509-3-5p, miR-508-3p, miR-514, miR-506, miR-513a-5p, miR-507, miR-509-3p, miR-513b, miR-876-3p, miR-378*, miR-935, miR-181a, miR-99a, miR-194, miR-592, miR-1296, miR-624*, miR-140-5p, miR-651) in monosomic compared to disomic tumors; eight miRNAs (miR-624, miR-509-3-5p, miR-508-3p, miR-506, miR-513a-5p, miR-509-3p, miR-513b, miR-935) were also differentially expressed in metastatic patients, all bearing chromosome 3 monosomy [221]. The study by Smit et al. showed the dysregulation of miRNAs in high-risk compared to low- and intermediate-risk tumors (upregulation of miR-132-5p, miR-151a-3p, miR-17-5p, miR-16-5p, and miR-21-5p, and downregulation of miR-181b-5p, miR-101-3p, miR-378d, miR-181a-2-3p, miR-99a-5p, let-7c-5p, miR-1537-3p, and miR-99a-3p) (see above Section 4 [202]). The association between miRNA expression and chromosomal aberration is still under investigation. Souri et al. investigated the association between miRNAs, HLA (human leukocyte antigen) expression, tumor-associated macrophages (TAMs), and TILs; the authors also evaluated chromosome 3 status, as it is known that it is correlated with inflammatory infiltrates, and *BAP1* expression. Two patterns of miRNAs have been identified, related to the up- or downregulation of HLAs and immune infiltrates, both related to chromosome 3 status and *BAP1* expression. These miRNAs may be considered potential therapeutic targets or inhibitors of inflammation [259].

By analyzing the UM dataset of TCGA, a Cox univariate regression analysis allowed to identify a signature of risk miRNAs including miR-195, miR-224, miR-365a, miR-365b, miR-452, miR-4709 and miR-7702, while miR-873 and miR-513c were considered protective miRNAs. This nine-miRNA signature showed good accuracy as a prognostic tool for UM in the TCGA cohort [238]. The same dataset was analyzed by Falzone et al. by stratifying tumor samples according to tumor stage (T3-T4 vs. T1-T2 or high- vs. low-grade) and patient status (deceased vs. alive) [239]. Expression analysis showed that seven top dysregulated miRNAs were common in both comparisons; in particular, miR-514a-3p, miR-508-3p, miR-509-3-5p, miR-513c-5p, and miR-513a-5p were downregulated, while miR-592 and miR-199a-5p showed increased levels in high- vs. low-grade tumors and deceased vs. alive patients. Dysregulated miRNAs showed a significant association with poor prognosis and worse OS, suggesting their prognostic value in UM [239]. A small subset of eight samples from the TCGA dataset was analyzed, comparing metastatic and non-metastatic patients. Six miRNAs identified with such analysis were subsequently validated in a cohort of forty-six patients: miR-592, miR-346, and miR-1247 were upregulated, while miR-506 and miR-513c showed decreased levels in metastatic tumors; differential expression of miR-196b was not validated. The correlation between miRNA expression and chromosome 3 status or *BAP1* expression showed: (i) a significant increase in miR-196b in patients with chromosome 3 monosomy and/or loss of *BAP1* expression; (ii) a significant correlation of miR-592 with chromosome 3 status, with increased expression in monosomic tumors with or without loss of *BAP1* expression [220]. Another report on metastasis-related miRNAs re-analyzed the entire TCGA cohort of eighty tumor samples, grouped according to the development of metastasis. This new analysis showed the differential expression of twenty-two miRNAs, three upregulated (miR-199a-5p, miR-708-5p, miR-592) and nineteen downregulated (miR-508-3p, miR-509-3p, miR-508-5p, miR-514a-3p, miR-506-3p, miR-509-3-5p, miR-513c-5p, miR-513a-5p, and miR-513b-5p) in metastatic tumors. Expression data were used to cluster miRNAs, suggesting a common regulatory mechanism sustaining their dysregulation in UM. Almost all the dysregulated miRNAs, except for one, were associated with OS [240]. Recently, Yu et al. used TCGA expression data to build a prognostic lncRNA–miRNA–mRNA competitive endogenous RNA (ceRNA) network by weighted gene co-expression network analysis [235]. Concerning miRNAs, univariate Cox proportional hazard regression identified significant 214 miRNAs; 1490 mRNAs and 199 lncRNAs were also identified as potential prognostic factors for UM. These data were used to build the prognostic ceRNA network, which included five mRNAs (*CGREF1*, *P4HA2*, *RGMB*, *PKNOX2*, *SLC6A6*), four miRNAs (miR-181b, miR-507, miR-548, miR-181a), and six lncRNAs (*PVT1*, *HCP5*, *EPB41L4A-AS1*, *BOLA3-AS1*, *SNHG7*, *GAS5*). By Kaplan–Meier survival analysis, low expression of the four miRNAs was associated with poor prognosis [235]. In the last year, this topic has been investigated. Sun et al. re-analyzed TCGA expression data and identified a signature including five miRNAs (miR-513a-5p, miR-506-3p, miR-508-3p, miR-140-3p, miR-103a-2-5p) representing an independent prognostic factor able to predict the prognosis of UM patients [216]. Another signature comprising three miRNAs (miR-1296, miR-199a, miR-508) was identified by using TCGA expression data and validated in two GEO datasets (GSE84976 [203] and GSE68828 [224]); the authors suggested the prognostic application of this signature in predicting OS [222].

Recently, the existence of miRNA isoforms, called isomiRs, has been reported [260]. Each miRNA may present several isoforms (even more than thirty different isomiRs), and the distinct isoforms may target dissimilar sets of RNA molecules [261,262]. Londin et al. re-analyzed the eighty tumor samples from TCGA to evaluate the expression of isomiRs and tRNA-derived fragments (tRFs) as prognostic factors in UM [263]. The authors described an UM-specific expression pattern of miRNAs, miRNA-arms, and isomiRs, showing that for miRNAs originating from 44% of miRNA-arms the most abundant isoform is not the “archetype” reported in miRbase (https://www.mirbase.org/, last accessed on 24 December 2022), stressing a misclassification of expression data and subsequent implications on miRNA targets. Moreover, thirty-two new miRNA loci exclusive or predominantly abundant in UM, which are the best candidate as diagnostic or prognostic biomarkers for the disease, were identified. By stratifying samples according to chromosome 3 status and *BAP1* mutations, reduced levels of the miR-508/514 miRNA cluster were observed in monosomic or *BAP1*-mutated patients; interestingly, this miRNA cluster maps on chromosome X, not 3. Furthermore, isomiRs from other loci were upregulated in monosomic or mutant patients (e.g., miR-199a/b). The opposite patterns of expression were observed when samples were stratified according to protective mutations on *SRSF2*/*SF3B1* or *EIF1AX*. Similarly, characteristic profiles of tRFs were reported, together with a correlation between tRF length and clinicopathological features. Differential expression of tRFs was reported in association with several clinical features. Specific isomiRs and tRFs were also associated with metastasis, showing differential expression in metastatic patients. These results show that both isomiRs and tRFs may be applied as diagnostic and prognostic biomarkers in UM [263].

#### 5.1.2. Molecular Functions of miRNAs in UM Tissues

Several studies focused on single miRNAs showing altered expression in UM patients, investigating their molecular function in UM cells. Reduced expression of miR-124a was observed in pairs of tumor and normal tissues from a small cohort of six UM patients; in vitro assays revealed that miR-124a ectopic expression inhibited cell proliferation, migration, and invasion of UM cells, also reducing tumor growth in vivo. The molecular function of miR-124a may be exerted through several targets, including *CDK4* (cyclin-dependent kinase 4), *CDK6* (cyclin-dependent kinase 6), *CCND2* (cyclin D2), and *EZH2* [219]. Lu et al. confirmed the downregulation of miR-124 and its role in the regulation of cell proliferation and invasion in vitro [218]. A recent paper reported the downregulation of miR-130a in a cohort of sixty-two UM patients and forty-two unaffected individuals; miR-130a levels were especially reduced in metastatic patients, and low miRNA expression was associated with shorter OS. Enforced overexpression of miR-130a impaired cell migration and invasion in vitro by inducing downregulation of *USP6* (ubiquitin specific peptidase 6) and inactivation of Wnt/β-catenin pathway; moreover, miRNA overexpression inhibited tumor growth in vivo [223]. Reduced levels of miR-137 were observed in tumors compared to normal tissues of UM patients; shorter OS and more adverse features were reported for patients with lower miRNA expression. Overexpression of miR-137 showed anti-tumor effects, decreasing cell proliferation, migration, and invasion in vitro by targeting *EZH2*, and inhibiting of Wnt/β-catenin pathway and epithelial-to-mesenchymal transition (EMT) [225]. MiRNA expression was evaluated in a cohort of twenty-three choroid samples: after a first microarray analysis performed in ten samples (five UM vs. five normal tissues), validation in real-time PCR (microarray samples plus six UM and seven normal tissues) confirmed the upregulation of miR-378d and miR-378g, and the downregulation of miR-204-5p, miR-143-3p, and miR-145-5p. Ectopic expression of miR-145-5p in UM cell lines reduced cell proliferation and increased apoptosis by targeting *IRS1* (insulin receptor substrate 1) [228]. The same group demonstrated that miR-145-5p tumor suppressive function was also performed through other targets, namely *NRAS* and *VEGFA* (vascular endothelial growth factor A), thus impairing invasion and angiogenesis in vitro, and also tumor growth and angiogenesis in vivo [230]. Lower expression of miR-145 and miR-205 was reported in both low- and high-invasive tumors compared to normal tissues. Ectopic expression of the two miRNAs (alone or in combination) impaired cell proliferation and invasion, with a stronger effect of miR-145 and miR-205, respectively. Both miRNAs synergistically regulated *CDC42* (cell division cycle 42) and *NRP1* (neuropilin 1 expression) [229]. MiR-155 showed increased levels in tumor tissues compared to paired normal tissues in a cohort of twenty-five UM patients; upregulation of miR-155 enhanced cell proliferation and invasion of UM cell lines through the target *NDFIP1* (Nedd4 family interacting protein 1) [233]. The expression of the miR-181 family was investigated in a small cohort of UM and normal tissues (three vs. three) by microarray, showing upregulation of mir-181b-1, mir-181b-2, and mir-181a; miR-181a and miR-181b upregulation was confirmed in a set of UM cell lines compared to the RPE (retinal pigment epithelium) cell line. Overexpression of miR-181b promoted cell cycle progression by regulating *CTDSPL* (CTD small phosphatase-like) expression [236]. Another study reported the downregulation of miR-182 in five out of seven UM clinical specimens, showing that its tumor suppressive function depended on *TP53* activation and regulated cell proliferation by targeting *MITF*, *BCL2* (BCL2 apoptosis regulator), and *CCND2*. In addition, overexpression of miR-182 inhibited tumor growth in vivo [237]. The *HDM2* (now named *MDM2*, MDM2 proto-oncogene) gene on 12q15 is highly expressed in 97% of UM [156]. It has been demonstrated that high *HDM2*/*MDM2* expression inhibits *TP53* and its function of removing abnormal cells [264]. *BCL2*, which is found on 18q21, is found to be highly expressed in both UM and normal melanocytes. This overexpression has been shown to inhibit apoptosis [93,264] and is thought to be responsible for melanocyte resistance to chemotherapy or irradiation [93,156]. Downregulation of miR-224-5p was observed in thirty pairs of tumor and normal tissues from UM patients. In vitro, increased expression of miR-224-5p reduced cell proliferation, migration, and invasion by regulating the expression of its targets *PIK3R3* (phosphoinositide-3-kinase regulatory subunit 3) and *AKT3* (AKT serine/threonine kinase 3) [244]. Reduced levels of miR-224-5p and regulation of cell proliferation, migration, and invasion were confirmed by Zheng et al. [245]. Zhou et al. investigated the expression of miR-20a, which was upregulated in ten UM tissues compared to ten normal ones [241]. The authors showed that miR-20a promoted cell proliferation, migration, and invasion of UM cell lines [241]. Reduced levels of miR-34a were reported by northern blot in three tumor tissues compared to normal ones; miR-34a negatively regulated cell proliferation and migration by targeting *MET* (MET proto-oncogene, receptor tyrosine kinase) and affecting the *AKT* pathway [251]. A few years later, the same group reported another target of miR-34a, namely *LGR4* (leucine-rich repeat-containing G protein-coupled receptor 4), through which the miRNA modulated cell migration, invasion, and EMT using *MMP2* (matrix metallopeptidase 2) as a downstream effector [252]. In contrast, upregulation of miR-34a, together with miR-146a and miR-21, was observed by our group in twelve FFPE tumors compared to choroidal melanocytes from five unaffected individuals [231]. Reduced expression of miR-34b and miR-34c was reported by Dong and Lou by analyzing five pairs of tumor and normal tissues [253]. In vitro assays showed that both miRNAs were involved in cell proliferation and migration by repressing *MET* expression. MiR-34b and miR-34c were also involved in sensitivity to doxorubicin treatment, and their expression was induced by the drug [253]. Comparing tumor and normal tissues from twenty-eight UM patients, Ling et al. reported the upregulation of miR-367 and its role in promoting proliferation and migration in vitro by targeting the well-known tumor suppressor *PTEN* (phosphatase and tensin homolog) [254]. Increased expression of miR-652 was shown in a cohort of twenty-six paired tumors and normal tissues; in vitro reduced miRNA expression impaired cell proliferation and migration through *HOXA9* (homeobox A9) upregulation, which is an inhibitor of the *HIF1A* (hypoxia-inducible factor 1 subunit alpha) signaling pathway [255].

Other studies investigated miRNA functions in vitro without evaluating their expression in patient biopsies. Let-7b expression was reduced in UM cells treated with radiation; increased let-7b expression enhanced radiosensitivity by targeting *CCND1*, thus promoting G1 arrest [215]. *CCND1* has been found to be overexpressed in 65% of UM cases. *CCND1* overexpression activates cyclin-dependent kinases (CDKs), which phosphorylate and inactivate the *RB1* protein [74,128]. Overexpression of *CCND1* is associated with large tumor size, epithelioid cytology, and poor prognosis [74]. Chen et al. reported a reduced expression of miR-137 in UM cell lines compared to primary melanocytes, coherently with the downregulation in tumor tissues reported by Zhang et al. (see above Section 5.1.2 [225]). Restoring miR-137 expression in UM cells caused cell cycle arrest by downregulating the targets *MITF* and *CDK6*, thus modulating several signaling pathways [226]. In another study, the role of miR-137 was also investigated in a few cancer models, including UM, where it regulated cell viability in vitro by downregulating *SRC3* (now renamed *NCOA3*, nuclear receptor coactivator 3). The same study also proved that miR-106b, miR-106a, miR-20a-5p, miR-519d, miR-17-5p, miR-429, miR-200c, miR-93-5p, and miR-372 suppressed *SRC3* expression and, except for miR-93-5p and miR-372, inhibited cell proliferation [217]. Recently, Quéméner et al. showed that miR-16 overexpression decreased cell proliferation and promoted apoptosis in UM cell lines, suggesting a tumor-suppressive function in UM [234]. The authors considered that the sole expression level of the miRNA is not useful to infer its activity because of the existence of ceRNA networks within cells. For this reason, the miR-16 interactome was investigated by RNA pull-down in UM cells compared to the same cells with ectopic miR-16 expression; downregulated genes were considered potential targets of miR-16, while upregulated genes were postulated to act as miRNA sponges. Sequence analysis showed that only 30% of downregulated miRNA targets and 2% of upregulated sponges contained miR-16-binding sites, suggesting a non-canonical base pairing. Indeed, a sequence motif found in the potential sponges suggested that the binding with miR-16 would create a bulge in the miRNA seed sequence, suppressing the miRNA-mediated silencing as expected according to the miRNA sponge hypothesis. In vitro assays showed that cell proliferation was impaired by decreased levels of miR-16 targets such as *AMOT* (angiomotin), *TACC1* (transforming acidic coiled-coil containing protein 1), *NRBP1* (nuclear receptor binding protein 1), and *DNAJB4* [DnaJ heat shock protein family (Hsp40) member B4]. Among the potential sponges, *PYGB* (glycogen phosphorylase B) showed the highest expression in UM and most increased levels after transfection with miR-16 mimic, both as mRNA and protein; non-canonical binding sites found in *PYGB* sequence may allow the miRNA sponge to interact with miR-16. The fifty-seven potential sponges of miR-16 showed a prognostic value in predicting survival in TCGA patients with unchanged levels of the miRNA. A signature related to miR-16 activity including four mRNAs (*TSPAN14*, *NLE1*, *FLNC*, and *LIPA*) was identified and validated in an independent cohort (GSE22138 [199]), suggesting a potential *application* in clinical practice [234]. Wang et al. observed an increased expression of *MMP2* and *MMP9* (matrix metallopeptidase 9) in choroidal malignant melanoma (CMM) compared to non-tumoral choroidal tissues; the authors demonstrated that both metalloproteases are targeted by miR-296-3p, which showed reduced expression in CMM cell lines compared to normal melanocytes. Ectopic expression of miR-296-3p inhibited cell proliferation, migration, and invasion and enhanced apoptosis in vitro [250]. The same study also reported that *MMP2* and *MMP9* expression was regulated by the lncRNA *FOXCUT* (FOXC1 upstream transcript); similar to miR-296-3p, *FOXCUT* increased expression, impaired cell proliferation, migration, and invasion, and promoted apoptosis in vitro; it also showed decreased levels in CMM cell lines compared to normal melanocytes. Not surprisingly, miR-296-3p and *FOXCUT* expressions were positively correlated [250]. Another study compared UM cell lines with normal tissues, reporting increased expression of miR-21, in agreement with miR-21 upregulation reported by our group in UM tissues (see above Section 5.1.2 [231]). In vitro assays showed that miR-21 overexpression promoted cell proliferation, migration, and invasion and reduced apoptosis by targeting *TP53*, while reduced expression of miR-21 impaired tumor growth in vivo [242]. Because of its involvement in EMT regulation, Wang et al. investigated the effects of miR-23a in UM cell lines: increased miRNA levels reduced *ZEB1* (zinc finger E-box binding homeobox 1) and *VIM* (vimentin) while enhancing *CDH1* (cadherin 1) expression; moreover, ectopic expression of miR-23a impaired the migration of UM cells [246]. Overexpression of *ZEB1* decreased the expression of miR-23a and *CDH1* while increasing *VIM* and *CDH2* (cadherin 2) levels. Hence, the authors reported a negative feedback loop involving miR-23a and *ZEB1* regulating EMT [246]. Guo et al. showed that increased expression of miR-26a inhibited cell viability and proliferation and promoted apoptosis [247]. These effects were mediated by the enhanced expression of *TP53* and the reduction in *MDM2* levels mediated by miRNA overexpression [247]. Shortly after, another paper confirmed the data about miR-26a; the authors reported that *EZH2*, which showed increased expression in UM tissues, was targeted by miR-26a, which in turn was downregulated in UM cell lines compared to normal choroidal cells. *EZH2* knockdown inhibited cell proliferation, such as miR-26a increased expression [248]. Reduced levels of miR-216a-5p promoted cell proliferation in vitro by targeting *HK2* (hexokinase 2), thus impairing glycolysis. Immunohistochemical staining of UM tissues showed a negative correlation of expression between miR-216a-5p and *HK2*; low miRNA levels were also associated with shorter OS and DFS [243]. It was reported that genistein, an isoflavone isolated from soybean with an antitumor effect, reduced miR-27a levels, which in turn affected *ZBTB10* (zinc finger and BTB domain containing 10) expression and reduced cell proliferation [249]. Liu et al. reported a negative correlation between miR-9 expression and invasion of UM cells; ectopic miRNA expression reduced migration and invasion in vitro by decreasing *NFKB1* (nuclear factor kappa B subunit 1) levels. By affecting *NFKB1* levels, miR-9 indirectly inhibited *MMP2*, *MMP9*, and *VEGFA* expression [256]. In vitro treatment of UM and CM cell lines with the histone deacetylase inhibitor MS-275 affected the expression of the miR-17-92a cluster in different ways, but always reduced miR-92a-3p. In vitro experiments showed that miR-92a-3p targeted *MYCBP2* (MYC binding protein 2), thus contributing to the apoptosis of UM cell lines. The authors also reported the upregulation of miR-92a-3p and the downregulation of *MYCBP2* in UM tissues, although the comparison was performed against normal skin samples [257].

Cancer stem cells (CSCs) play a crucial role in cancer progression. Joshi et al. showed that distinct UM cell lines had a different sensitivity to natural killer (NK) cell-mediated cytolysis because of the expression and secretion of specific miRNAs regulating NK cells, namely miR-155 and miR-146a [232].

### 5.2. LncRNAs Regulate Epigenetic Mechanisms in UM

Another class of ncRNAs that has been discovered more recently is represented by lncRNAs, which are similar to mRNAs in size, structure, transcriptional regulation, and post-translational modifications. Indeed, lncRNAs are transcribed by RNA polymerase II, may contain introns, and are capped, spliced, and polyadenylated. Their size is heterogeneous, ranging from 200 nts to kilobases [265]. Unlike miRNAs, it has been demonstrated that lncRNAs may play several different functions, and only for a few of them has the precise mechanism of action been described [266]. One of the most commonly investigated functions is the participation in ceRNA networks, which causes the modulation of miRNA function on target mRNAs. The ceRNA hypothesis has been investigated in several tumor models, including UM. Our group recently developed an online tool to build ceRNA networks including lncRNAs, miRNAs, and mRNAs, in UM using TCGA expression data [267]. In recent years, lncRNAs have gained increasing attention as epigenetic regulators both in physiological and pathological processes. Accordingly, in the context of UM, a growing literature is available. Similar to miRNAs and mRNAs, lncRNAs have also been investigated as prognostic biomarkers in UM (Table 3).

#### 5.2.1. LncRNAs Promote UM Progression

The first study investigating the role of lncRNA in UM carcinogenesis reported the downregulation of *PAUPAR* (PAX6 upstream antisense RNA) in twelve UM compared to five unaffected tissues. Overexpression of the lncRNA impaired colony formation ability and migration in vitro, as well as tumor growth in vivo. A potential downstream effector of *PAUPAR* was *HES1* (hes family bHLH transcription factor 1), whose expression was higher in UM tissues and decreased after in vitro ectopic expression of the lncRNA. *HES1* increased levels promoted migration of UM cells and were sustained by the inhibition of histone H3K4 methylation induced by *PAUPAR*, which localized within the nucleus of UM cells [287]. The role of *ANRIL*, now named *CDKN2B-AS1* (CDKN2B antisense RNA 1), was investigated in both CM and UM by Pan et al. [275]: by analyzing CM and UM tissues compared to a unique cohort of normal tissues from both skin and choroid, the authors showed the increased expression of *ANRIL*/*CDKN2B-AS1* in both diseases. In parallel, the expression of tumor suppressive proteins coded by the same locus but in a different direction was analyzed, showing a significant downregulation of *INK4A* (p16, now named *CDKN2A*) and *INK4B* (p15, now named *CDKN2B*, cyclin-dependent kinase inhibitor 2B), while *ARF* (p14, now considered a splicing variant of *CDKN2A*) levels were unchanged. The transient silencing of *ANRIL*/*CDKN2B-AS1* triggered an increase in *INK4A*/*CDKN2A* and *INK4B*/*CDKN2B* expression and impaired the metastatic ability of CM and UM cells in vitro; also, tumor formation induced by UM cells was inhibited both in vitro and in vivo by lncRNA knockdown [275]. Upregulation of *MALAT1* (metastasis-associated lung adenocarcinoma transcript 1), a well-known oncogene in several cancer models, was also reported in twenty-five UM tissues compared to their normal counterparts; decreased expression of *MALAT1* reduced cell proliferation, colony formation, migration, and invasion in vitro. *MALAT1* downregulation induced increased expression of *PCNA* (proliferating cell nuclear antigen), Ki-67/*MKI67*, and miR-140-3p; downregulation of miR-140-3p was also confirmed in UM tissues [227]. Wu et al. also demonstrated that *MALAT1* regulated *HOXC4* (homeobox C4) expression by binding miR-608, thus suppressing tumor growth in vivo [285]. *RHPN1-AS1* [RHPN1 antisense RNA 1 (head to head)] is a cytoplasmic lncRNA upregulated in UM tissues; its knockdown inhibited cell proliferation, migration, and invasion in vitro and also tumor growth in vivo [291]. *CASC15* (cancer susceptibility 15) is an lncRNA involved in many tumors. Xing et al. identified a new isoform of *CASC15*, which they named CASC15-NewTranscript 1 (CASC15-NT1 or CANT1), and reported decreased levels of this new lncRNA in seventeen UM patients compared to twelve normal individuals [277]. Overexpression of CASC15-NT1/CANT1 impaired cell migration and colony formation ability in vitro and reduced tumor growth in vivo. A potential downstream effector of CASC15-NT1/CANT1 was identified in *XIST* (X inactive specific transcript): indeed, *XIST* levels increased after induction of CASC15-NT1/CANT1 expression in UM cells derived from female patients through the transcriptional activation of other lncRNAs, namely *JPX* (JPX transcript, XIST activator) and *FTX* (FTX transcript, XIST regulator). CASC15-NT1/CANT1 directly bound the promoter of *JPX* and *FTX* loci, inducing histone H3K4 methylation [277]. Upregulation of *HOXA11-AS* (HOXA11 antisense RNA) was observed in five UM compared to matched control tissues, while its reduced expression inhibited cell proliferation and promoted apoptosis. *HOXA11-AS* was prevalently localized within the nucleus, where it interacted with *EZH2* and induced silencing of p21/*CDKN1A* (cyclin-dependent kinase inhibitor 1A); in the cytoplasm, *HOXA11-AS* acted as a sponge for miR-124, which was downregulated in UM tissues. Ectopic expression of miR-124 impaired cell proliferation and invasion (as previously discussed) [218]. *FTH1P3* (ferritin heavy chain 1 pseudogene 3) showed increased expression in a cohort of twenty-five tumor samples compared to non-tumor matched tissues; this upregulation sustained increased cell proliferation and migration in vitro, promoted *RAC1* (Rac family small GTPase 1) and Frizzle 5 (*FDZ5*, frizzled class receptor 5) expression, and induced downregulation of miR-224-5p, which in turn reduced the expression of the lncRNA in a loop regulative mechanism. MiR-224-5p was downregulated in UM tissues (coherently with Li et al. [244]) and mediated the *FTH1P3*-induced effects on proliferation and migration [245]. *PVT1* (Pvt1 oncogene) upregulation in forty UM patients was reported by Huang et al. [288]. Knockdown of *PVT1* reduced cell proliferation and colony formation ability and increased apoptosis in vitro by downregulating *EZH2* [288]. Another study confirmed *PVT1* upregulation and reported its role in promoting cell proliferation, migration, and invasion in vitro; the authors demonstrated the binding between *PVT1* and miR-17-3p in the cytoplasm, which caused the increased expression of *MDM2* and the consequent degradation of *TP53* [289]. *ZNNT1* (ZNF706 neighboring transcript 1), downregulated in twenty UM tissues, showed a prevalent localization within the nucleus and regulated the expression of *ATG12* (autophagy-related 12), thus promoting autophagy of UM cells in vitro. *ZNNT1* overexpression also increased cell death and impaired migration in vitro and inhibited tumor growth in vivo [297]. In a cohort of eight enucleated eyes and five normal eyes, *GAS5* (growth arrest-specific 5) reduced expression was reported and associated with poor prognosis. Knockdown and ectopic expression of *GAS5* showed its role in inhibiting cell viability, migration, invasion, and EMT in vitro, performed through the sponging of miR-21 [280]. Previously discussed papers showed the upregulation of miR-21, consistent with these results (see above Section 5.1.2 [231,242]). Recently, our group reported the increased expression of *LINC00518* (long intergenic non-protein coding RNA 518) and *LINC00634*, now named *SMIM45* (small integral membrane protein 45), in a cohort of forty-one UM patients comparing tumor and normal adjacent tissues. In vitro experiments in UM cells showed that *LINC00518* downregulation impaired cell proliferation and migration, while its expression was increased by triggering EMT and hypoxia-like response. We proposed that *LINC00518* molecular effectors may be five mRNAs, namely *LINGO2* (leucine-rich repeat and Ig domain containing 2), *NFIA* (nuclear factor I A), *OTUD7B* (OTU deubiquitinase 7B), *SEC22C* (SEC22 homolog C, vesicle trafficking protein), and *VAMP3* (vesicle-associated membrane protein 3), showing decreased levels after lncRNA transient silencing. *LINC00518* may regulate mRNA expression by acting as an miRNA sponge or by directly binding the 3′-untranslated region (UTR) of mRNAs, thus masking miRNA-binding sites and acting as an “miRNA protector”. Finally, our data suggested *MITF* as a potential regulator of *LINC00518* expression [281]. By analyzing tumor-normal pairs from three UM patients, Li et al. identified a new isoform of the lncRNA *LOC100505912* (uncharacterized LOC100505912), named Oncotarget in UM formation-transcript 1 (OUM1), showing increased expression in tumor tissues [250]. Knockdown of its expression decreased cell viability, proliferation, migration, and invasion in vitro, while tumor growth was impaired in vivo. *LOC100505912*/OUM1 reduced the expression of *PTPRZ1* (protein tyrosine phosphatase receptor type Z1), which was upregulated in UM tissues. *LOC100505912*/OUM1, mainly localized in the cytoplasm, was able to directly bind *PTPRZ1*, thus enhancing protein tyrosine phosphatase (*PTP*) activity. Since *PTP* is involved in chemoresistance, nanoparticles for the delivery of siRNAs against *LOC100505912*/OUM1 or *PTPRZ1* were tested, showing a successful reduction in tumor growth and pulmonary metastasis in vivo [283]. As previously discussed, Wang et al. reported the downregulation of *FOXCUT* in CMM, showing its role in modulating cell proliferation, apoptosis, migration, and invasion in vitro by regulating *MMP2* and *MMP9* expression [250].

Also for lncRNAs, reports discussing their function in UM without appropriately investigating their expression in patient biopsies are available. Downregulation of the transcription factor *HIC1* (HIC ZBTB transcriptional repressor 1) was observed in three UM samples compared to three normal tissues and in UM cell lines compared to normal dermal fibroblasts [284]. Overexpression of *HIC1* suppressed cell proliferation, colony formation, and invasion; moreover, *HIC1* altered the expression of seventy-six lncRNAs, among which *LOC101928143* (uncharacterized LOC101928143) showed increased expression. Because of the position of this locus upstream of the protein-coding gene *NUMB* (NUMB endocytic adaptor protein), the authors called this dysregulated transcript lncRNA-numb; in particular, the 635 nt isoform was downregulated in UM cells compared to dermal fibroblasts. Ectopic expression of *LOC101928143* inhibited cell proliferation, colony formation, and invasion of UM cells, suggesting that it may act as a downstream effector of *HIC1* [284]. Pan et al. reported the overexpression of *P2RX7* (purinergic receptor P2X 7) variant 3, called P2RX7-V3 (NR_033949.1), in invasive UM cell lines compared to ARPE-19, a human RPE cell line. Knockdown of this non-coding variant inhibited the migration and colony formation ability of UM cells in vitro. Decreased levels of P2RX7-V3 in vivo reduced the expression of Ki-67/*MKI67* and *VIM* and increased *CDH1* levels, repressing tumor growth and progression [286]. Expression modulation of twenty lncRNAs has been reported after the treatment of UM cells with olaparib, a poly-(ADP-ribose) polymerase (*PARP*) inhibitor. Investigation of PanCancer samples showed that these lncRNAs are often genetically altered in cancer and that some of these genetic alterations were associated with OS [298].

#### 5.2.2. LncRNAs with Prognostic Value in UM

Concerning the potential application of lncRNAs as prognostic biomarkers, Xu et al. performed a transcriptome analysis in eleven UM tumors (six metastasized and five non-metastasized), identifying 329 differentially expressed lncRNAs (DELs) and 802 differentially expressed mRNAs (DEMs). *RP1-272L16.1* and *RP11-329N22.1* were the most significantly up and downregulated DELs. The authors showed that lncRNA deregulation was coherent with chromosomal status: indeed, DELs mapping on chromosomes 3 and 8 was down- or upregulated according to chromosome 3 loss and chromosome 8 gain. The dysregulation of twelve DELs randomly selected was confirmed in real-time PCR [292]. *PVT1*, which plays oncogenic functions in several cancer models, was investigated in the TCGA dataset for its prognostic value. Increased levels of *PVT1* were associated with malignant features of UM (older age, epithelioid morphology, distant metastasis, extrascleral extension, higher death rate) and predicted poor OS; this predictive value was specific to UM and was not observed in CM. Upregulation of *PVT1* in UM was caused by DNA amplification and methylation [290] and was also observed in UM tissues (see above Section 5.2.1 [288,289]). A previously discussed analysis of the TCGA dataset identified forty-seven CpG islands of lncRNAs with aberrant methylation in patients grouped according to survival time. A nine-lncRNAs-CpG-classifier with prognostic value for patient survival was identified (see above Section 4 [205]). In the TCGA dataset, decreased levels of *SNHG7* (small nucleolar RNA host gene 7) were associated with poor prognosis, with significant downregulation of the lncRNA in metastatic patients compared to non-metastatic patients. Ectopic expression in UM cell lines promoted cell cycle arrest and apoptosis in vitro, potentially through *EZH2* as a downstream effector [295]. The same dataset allowed Wu et al. to identify *SNHG15* (small nucleolar RNA host gene 15) as a prognostic factor for death in UM: indeed, high levels of *SNHG15* were associated with worse prognosis, pathologic state, and metastasis [294]. *LOC100132707*, now named *PAXIP1-AS2* (PAXIP1 antisense RNA 2), was upregulated in tumors from metastatic patients in the TCGA cohort. In vitro assays showed it promoted cell migration and invasion by upregulating *JAK2* (Janus kinase 2), while reduced expression of *LOC100132707*/*PAXIP1-AS2* impaired tumorigenesis in vivo [282]. By re-analyzing expression data from the GSE22138 dataset [199], Yang et al. reported decreased levels of *ZNF667-AS1* [ZNF667 antisense RNA 1 (head to head)] in metastatic patients compared to non-metastatic UM patients; follow-up data from TCGA showed the association of the lncRNA with histological type, metastasis, recurrence, and death, with a poorer prognosis in the low expression group. Among the correlated genes, *MEGF10* (multiple EGF-like domains 10) showed a high positive expression correlation and a prognostic value consistent with *ZNF667-AS1*. In vitro experiments showed that both *ZNF667-AS1* and *MEGF10* modulated cell viability, cell cycle progression, and apoptosis in UM cells [296]. A signature of six autophagy-related lncRNAs (*AL589843.1*, *AC016757.1*, *LINC00957*, *AP005121.1*, *PVT1*, *AC016747.1*) with prognostic value was identified in TGCA and GSE22138 [199] by comparing metastatic and non-metastatic patients [269]. Another signature of autophagy-related lncRNAs was reported by Chen et al.; six lncRNAs (*SOS1-IT1*, *AC016747.1*, *AC100791.3*, *AC018904.1*, *AC104825.1*, *AC090617.5*) were prognostic for OS in UM patients both individually and in combination [270]. Focusing on the post-transcriptional modification N6-methyladenosine (m6A), Liu et al. evaluated the prognostic value of lncRNAs related to m6A [268]. First, they investigated the expression of m6A regulators in TGCA, reporting a prognostic value for a signature including *RBM15B* (RNA binding motif protein 15B), *YTHDF3* (YTH N6-methyladenosine RNA binding protein 3), and *IGF2BP2* (insulin-like growth factor 2 mRNA binding protein 2), and their association with immune infiltration in UM. By correlation analysis, a signature of thirty-eight lncRNAs correlated to the three regulators was identified; the expression of the lncRNAs allowed to classify TCGA samples into two clusters differing in immune infiltrate, OS, and progression-free survival. Five lncRNAs (*AC008555.4*, *AC018529.1*, *AC104129.1*, *CYTOR*, *MIR4435-2HG*) were associated with the prognosis of UM patients. Thirteen immune-related genes showing expression correlation with five lncRNAs and associated with poor prognosis showed differential expression in the two clusters, suggesting they may be downstream effectors of m6A regulators and lncRNAs [268]. Classification of TCGA samples according to immune infiltrate led to the identification of eight immune-related lncRNAs (*ZNF667-AS1*, *ZNF350-AS1*, *LINC02572*, *LINC02367*, *ACVR2B-AS1*, *DKFZP434A062*, *CYTOR*, *LINC01615*) with prognostic value. The authors also suggested a possible application in therapy by identifying potential target genes and specific drugs [274]. Increased expression of *SAMMSON* (survival-associated mitochondrial melanoma-specific oncogenic non-coding RNA) was observed in metastatic patients from the TCGA cohort. Knockdown of the lncRNA impaired cell viability and induced apoptosis in UM cell lines and suppressed tumor growth in vivo. Several interactors of *SAMMSON* were identified, including p32 (*C1QBP*, complement C1q binding protein) and *XRN2* (5′-3′ exoribonuclease 2); the lncRNA interactome affected mitochondrial function and protein synthesis [293]. Wang et al. investigated the expression, mutations, and prognostic value of *BACE1-AS* (BACE1 antisense RNA) in thirty-two cancer models from TCGA: higher expression levels of *BACE1-AS* were associated with a longer OS and progression-free interval in the TCGA UM dataset [276]. Recently, the role of lncRNAs in the regulation of ferroptosis was investigated. A signature of five lncRNAs (*AC136475.3*, *AC104129.1*, *PPP1R14B-AS1*, *LINC00963*, *ZNF667-AS1*) related to ferroptosis was identified, showing a predictive value for mortality risk; tumors classified according to the lncRNA signatures also showed differences in the immune response. Three lncRNAs were further studied through in vitro assays, namely *LINC00963* (long intergenic non-protein coding RNA 963), *PPP1R14B-AS1* (PPP1R14B antisense RNA 1), and *ZNF667-AS1*. Knockdown of the three lncRNAs impaired cell migration and invasion; effects on cell proliferation were different for each tested lncRNA [272]. LncRNAs related to fatty acid metabolism (FAM) were identified by correlation with FAM-related mRNAs identified in TCGA expression data. Twenty-five FAM-related lncRNAs with prognostic value were identified: among them, *AC104129.1*, *SOS1-IT1* (SOS1 intronic transcript 1), and *DLGAP1AS2* (DLGAP1 antisense RNA 2) were associated with high-risk, while *IDI2-AS1* (IDI2 antisense RNA 1) with low-risk [273]. A recent analysis of the TCGA dataset allowed to identify 1664 immune-related genes and 2216 lncRNAs thanks to expression correlation analysis. Through regression analysis, a signature of three lncRNAs (*AP005121.1*, *AC104117.3*, and *SOX1-OT*) able to predict OS was identified [271]. As previously discussed, Yu et al. used TCGA expression data to identify a prognostic signature including five mRNAs (*CGREF1*, *P4HA2*, *RGMB*, *PKNOX2*, *SLC6A6*), four miRNAs (miR-181b, miR-507, miR-548, miR-181a) and six lncRNAs (*PVT1*, *HCP5*, *EPB41L4A-AS1*, *BOLA3-AS1*, *SNHG7*, *GAS5*). Kaplan–Meier survival analysis showed that low expression of *EPB41L4A-AS1*, *BOLA3-AS1*, *SNHG7*, and high levels of *PVT1* were associated with low survival rates [235].

#### 5.2.3. CircRNA Involvement in UM

A particular class of lncRNAs is represented by circRNAs, including molecules with the peculiar characteristic of the covalent binding between the two ends achieved through a specific splicing mechanism, defined as backsplicing [299]. As well as the other classes of ncRNAs, circRNAs act as epigenetic regulators and have been associated with several diseases, including cancer. To date, only a few data about circRNA involvement in UM are available. The first study was published in 2018 by Yang et al., who performed microarray analysis in five UM tissues and five normal samples [278]. The microarray allowed to detect the expression of 170,340 circRNAs, among which 50,579 were differentially expressed in tumor tissues. Fifteen circRNAs were validated in twenty UM and twenty normal tissues, confirming the upregulation of seven circRNAs (circ_0119873, circ_0128533, circ_0047924, circ_0103232, and circRNA10628-6) and the downregulation of two (circ_0032148 and circ_0133460). Since circRNAs may act as miRNA sponges, the authors identified the potential miRNAs targeted by the dysregulated circRNAs, which included miR-145, miR-92a-3p, miR-193a, miR-193b, and miR-204 [278]. Reduced expression of miR-145 and miR-204 and upregulation of miR-92a-3p in UM have been previously discussed (see above [228,257]). By analyzing microarray data of this first study about circRNAs, Liu et al. identified circ_0119872, transcribed from exons 4 and 5 of RASGRP3, as upregulated in tumor tissues [279]. Circ_0119872 promoted cell proliferation, migration, invasion, and angiogenesis both in vitro and in vivo. The circRNA was mainly localized in the cytoplasm, where it directly interacted with miR-622, which in turn targeted *G3BP1* (G3BP stress granule assembly factor 1). *G3BP1* was upregulated in metastatic compared to non-metastatic tumors from the GSE44295 dataset and positively regulated cell proliferation and angiogenesis in vitro [279].

## 6. Conclusions

UM generally shows a low mutational burden, and only a few mutually exclusive driver mutations and chromosomal abnormalities with relevant diagnostic/prognostic significance occur in patients. Although the primary oncogenic processes in UM are well understood, a greater comprehension of their succession and biological repercussions is required to reveal the evolution of UM and, accordingly, successfully prevent or treat metastases. Furthermore, the few genetic anomalies cause several dozen heterogeneous molecular alterations reported in UM both in vitro and in vivo, influencing many aspects of the tumor and being related to the prognostic features of patients. These findings depict a concealed complex scenario that combines some homogeneous genetic changes with a cascade of disparate epigenetic dysregulations inducing melanocyte transformation and UM dissemination.

To date, these epigenetic alterations, particularly those involving ncRNAs, are being studied in order to discover both therapeutic molecules that can be targeted using RNA-based therapies and new diagnostic and prognostic markers that can be measured in biological fluids in a non-invasive manner [214,267,300].

## Figures and Tables

**Table 1 cancers-15-00775-t001:** Mutations selected from literature and verified by COSMIC (https://cancer.sanger.ac.uk/cosmic; last accessed on 28 December 2022). “c.?”: the nucleotide change could not be identified. Most frequent mutations in UM are highlighted in bold; no *BAP1* hotspot mutations have been described in literature.

Gene	Chr	Gene Function	Mutation Nucleotide Change	Functional Change	Ref
** *GNAQ* **	9q21.2	Mediating signaling between G-protein-coupled receptors and downstream effectors and upregulating MAPK pathway	Exon 4		
			A>T	p.T96S	[68,69,70,71]
			C>T	p.P170S	
			c.?	p.Q176R	
			c.?	p.R183C	
			c.?	p.R183H	
			c.?	p.I189T	
			c.?	p.P193L	
			A>G	Y192C	
			T>C	F194L	
			C>T	P170S	
			Exon 5		
			c.?	p.M203V	
			**A>T**	**p.Q209L**	
			**A>C**	**p.Q209P**	
			c.?	p.Q209M	
			**A>C**	**p.Q209H**	
			c.?	p.Q209I	
			c.?	p.F228L	
			G>A	D236N	
			C>T	L232F	
			T>C	V230A	
			G>A	M227I	
			c.?	p.V344M	
** *GNA11* **	19p13.3	Mediating signaling between G-protein-coupled receptors and downstream effectors and upregulating MAPK pathway	Exon 2		
			c.?	p.G48L	[68,69,70,71]
			exon 4		
			G>A	p.R166H	
			C>T	R166C	
			T>C	I200T	
			**C>T**	**p.R183C**	
			c.?	p.R183H	
			Exon 5		
			**A>T**	**p.Q209L**	
			C>T	S225F	
			G>A	V206M	
			**A>C**	**p.Q209P**	
			c.?	p.Q209Y	
			c.?	p.E221D	
			c.?	p.E234K	
			exon 7		
			c.?	p.R338H	
** *BAP1* **	3p21.1	Deubiquitinating hydrolase involved in tumor suppressor activity, DNA damage response, and proliferation	C>T	Q441*pe	[69,71,72,73]
			C>T	Q40*	
			A >T	E685V	
			del/insAGAG	Q456Rfs*115	
			C>G	Y33*	
			C>G	D68G	
			A>G	G185R	
			G>C	Q684*	
			C>T		
			insT	F170Lfs*13	
			T>A	p.Q590L	
			A>C	p.L101R	
			delATTCATCTTCCCGCGGGGCGGCCCCTCAGCGCCATGTCC	Removal of start site	
			delG	p.F50LfsX22	
			delAGGGCCCT	Deletion of splice donor and 6 base pair of exon	
			delCT	p.R300GfsX6	
			C>G; delA		
			C>A	p.R146M	
			delAGCACCAGCGGGGACTTGTTG	p.S289RfsX41	
			C>A	p.E007*	
			delGGCTGCTGGACCCCTGGCTGCCTTGGATTGGTCTGATGGA	p. S585Qfs*19	
			T>C	p.D68G	
			delTGTGAGCCAGGATGAAGGCACTGCAGCCTACCTCAGGGCT-GAAACCCTTG GTGAAGTCCTTCATGCGACTCAGGGTGGGTCCCAGGTCCAC-GCTGCTGCA GTTCAGGAGCACGCTCAGCAAGGCATGAGTTGCACAAGAGTTGGGTATCAG	p.L86_E125del	
			A>C	p.Y401*	
			delA	p.L262Rfs*2	
			delC	p.L186*	
			G>C	G8R	
			G>C	G9R	
			G>A	G9D	
			G>A	G15D	
			G>A	G6D	
			T>A	W70R	
			20_22del	K7_G8delinsR	
			C>T	P2L	
** *EIF1AX* **	Xp22.12	Involved in eukaryotic translation initiation	C>A	p.G9V	[69,71,72]
			C>G	p.G9R	
			C>A	p.G9V	
			C>T	p.G9D	
			T>C	p.K7R	
			T>C	Splice acceptor	
			C>T	p.G9D	
			C>T	p.G8E	
			C>G	p.G9R	
			G>C	G8R	
			G>C	G9R	
			G>A	G9D	
			G>A	G15D	
			G>A	G6D	
			T>A	W70R	
			C>T	P2L	
			c.20_22del	K7_G8delinsR	
** *SF3B1* **	2q33.1	Essential for splicing	**G>A**	**R625H**	[69,71]
			**C>T**	**R625C**	
			A>C	K666T	
			A>G	H662R	
			A>C	T663P	
** *SRSF2* **	17q25.1	Essential for splicing	c.274_300del	Y92_H100del	[69,71]
			c.274_297del	Y92_H99del	
			c.519_536del	S174_S179del	
** *PLCB4* **	20p12.3	Important role in the intracellular transduction of many extracellular signals in the retina.	G>T	D630Y	[69,71]
			G>A	D630N	
			G>T	D630V	
** *TERT* **	5p15.33	Telomerase reverse transcriptase activity	C>T	Increases the likelihood of the sequence to bind ETS from 78.4 (wild type) to 86.3 (mutation)	[71,74]
** *CYSLTR2* **	13q14.2	Involved in immune response	T>A	L129Q	[69,71]

**Table 2 cancers-15-00775-t002:** Dysregulated miRNAs in UM. NA (not available): it was impossible to univocally identify the mature miRNA either from the cited paper or miRBase.

miRNA	miRBase ID	Dysregulation	Function	Targets	Ref
let-7b	hsa-let-7b-5p	Upregulated in high-risk tumors			[201]
		Downregulated in UM cells treated with radiations	Regulation of radiosensitivity and cell cycle arrest	CCND1	[215]
let-7c-5p	let-7c-5p	Downregulated in high-risk tumors			[202]
miR-101-3p	hsa-miR-101-3p	Downregulated in high-risk tumors			[202]
miR-103a-2-5p	hsa-miR-103a-2-5p				[216]
miR-106a	hsa-miR-106a-5p		Regulation of cell proliferation in vitro	SRC3/NCOA3	[217]
miR-106b	hsa-miR-106b-5p		Regulation of cell proliferation in vitro	SRC3/NCOA3	[217]
miR-124	hsa-miR-124-3p	Downregulated in tumor tissues	Regulation of cell proliferation and invasion in vitro		[218]
miR-124a		Downregulated in tumor tissues	Regulation of cell proliferation, migration, and invasion in vitro and tumor growth in vivo	CDK4, CDK6, CCND2, EZH2	[219]
miR-1247	hsa-miR-1247-5p	Upregulated in metastatic tumors			[220]
miR-1296	hsa-miR-1296-5p	Downregulated in tumors with chromosome 3 monosomy			[221]
		Downregulated in advanced tumors			[222]
miR-130a	hsa-miR-130a-3p	downregulated in tumor tissues, especially in metastatic patients; low levels associated with shorter overall survival	Regulation of cell migration and invasion in vitro and tumor growth in vivo	USP6	[223]
miR-132-5p	hsa-miR-132-5p	Upregulated in high-risk tumors			[202]
miR-134	hsa-miR-134-5p	Upregulated in tumors with chromosome 3 monosomy; associated with liver metastasis			[224]
miR-135a*	hsa-miR-135a-5p	Upregulated in tumors with chromosome 3 monosomy			[221]
miR-137	NA		Regulation of cell proliferation, migration, invasion and EMT in vitro	EZH2	[225]
	hsa-miR-137-3p	Downregulated in UM cell lines compared to primary melanocytes	Regulation of cell cycle arrest	CDK6, MITF	[226]
	hsa-miR-137-3p		Regulation of cell viability in vitro	SRC3/NCOA3	[217]
miR-140-3p	hsa-miR-140-3p	High levels associated with better survival outcomes			[216]
		Downregulated in tumor tissues			[227]
miR-140-5p	hsa-miR-140-5p	Downregulated in tumors with chromosome 3 monosomy			[221]
miR-142-5p	hsa-miR-142-5p	Upregulated in tumors with chromosome 3 monosomy			[221]
miR-143	hsa-miR-143-3p	Upregulated in high-risk tumors			[201]
		Upregulated in tumors with chromosome 3 monosomy			[224]
miR-143-3p		Downregulated in tumor tissues			[228]
miR-145	hsa-miR-145-5p	Downregulated in tumor tissues	Regulation of cell proliferation and invasion	CDC42, NRP1	[229]
miR-145-5p		Downregulated in tumor tissues	Regulation of cell proliferation and apoptosis in vitro	IRS1	[228]
			Regulation of invasion and angiogenesis in vitro and tumor growth and angiogenesis in vivo	NRAS, VEGFA	[230]
miR-146a	hsa-miR-146a-5p	Upregulated in tumor tissues			[231]
		associated with different sensitivity to natural killer (NK) cells-mediated cytolysis			[232]
miR-146b	hsa-miR-146b-5p	Upregulated in tumors with chromosome 3 monosomy			[224]
miR-149*	hsa-miR-149-3p	Associated with liver metastasis			[224]
miR-151a-3p	hsa-miR-151a-3p	Upregulated in high-risk tumors			[202]
miR-1537-3p	hsa-miR-1537-3p	Downregulated in high-risk tumors			[202]
miR-155	hsa-miR-155-5p	Upregulated in tumor tissues	Regulation of cell proliferation and invasion	NDFIP1	[233]
		Associated with different sensitivity to natural killer (NK) cells-mediated cytolysis			[232]
miR-16	hsa-miR-16-5p	Tumor suppressive function	Regulation of cell proliferation and apoptosis in vitro	AMOT, TACC1, NRBP1, DNAJB4; sponged by PYGB	[234]
miR-16-5p		Upregulated in high-risk tumors			[202]
miR-17-5p	hsa-miR-17-5p	Upregulated in high-risk tumors			[202]
			Regulation of cell proliferation in vitro	SRC3/NCOA3	[217]
miR-181a	hsa-miR-181a-5p	Downregulated in tumors with chromosome 3 monosomy			[221]
		Low expression associated with poor prognosis			[235]
		Upregulated in UM cell lines UM cell lines compared to the RPE cell line			[236]
mir-181a	premiR	Upregulated in tumor tissues			[236]
miR-181a-2-3p	hsa-miR-181a-2-3p	Downregulated in high-risk tumors			[202]
miR-181b	hsa-miR-181b-5p	Low expression associated with poor prognosis			[235]
		Upregulated in UM cell lines UM cell lines compared to the RPE cell line	Regulation of cell cycle progression in vitro	CTDSPL	[236]
mir-181b-1	premiR	Upregulated in tumor tissues			[236]
mir-181b-2	premiR	Upregulated in tumor tissues			[236]
miR-181b-5p	hsa-miR-181b-5p	Downregulated in high-risk tumors			[202]
miR-182	hsa-miR-182-5p	Downregulated in tumor tissues	TP53-dependent regulation of cell proliferation; regulation of tumor growth in vivo	BCL2, CCND2, MITF	[237]
miR-193b	hsa-miR-193b-3p	Upregulated in high-risk tumors			[201]
miR-194	hsa-miR-194-5p	Downregulated in tumors with chromosome 3 monosomy			[221]
miR-195	hsa-miR-195-5p	Associated with high risk			[238]
miR-196b	hsa-miR-196b-5p	Upregulated in patients with chromosome 3 monosomy and/or loss of BAP1 expression			[220]
miR-199a	hsa-miR-199a-5p	Upregulated in high-risk tumors			[201]
		Upregulated in advanced tumors			[222]
		Upregulated in tumors with chromosome 3 monosomy			[224]
miR-199a-5p		Upregulated in high- vs. low-grade tumors and deceased vs. alive patients associated with poor prognosis and worse overall survival			[239]
		Upregulated in metastatic tumors			[240]
miR-199a*	hsa-miR-199a-3p	Upregulated in high-risk tumors			[201]
miR-200c	hsa-miR-200c-3p		Regulation of cell proliferation in vitro	SRC3/NCOA3	[217]
miR-204-5p	hsa-miR-204-5p	Downregulated in tumor tissues			[228]
miR-205	hsa-miR-205-5p	Downregulated in tumor tissues	Regulation of cell proliferation and invasion	CDC42, NRP1	[229]
miR-20a	hsa-miR-20a-5p	Upregulated in tumor tissues	Regulation of cell proliferation, migration, and invasion in vitro		[241]
miR-20a-5p			regulation of cell proliferation in vitro	SRC3/NCOA3	[217]
miR-21	hsa-miR-21-5p	Upregulated in tumor tissues			[231]
		Upregulated in UM cell lines compared with normal tissues	Regulation of cell proliferation, migration, invasion, and apoptosis in vitro and tumor growth in vivo	TP53	[242]
miR-21-5p		Upregulated in high-risk tumors			[202]
miR-214	hsa-miR-214-3p	Upregulated in tumors with chromosome 3 monosomy			[224]
miR-216a-5p	hsa-miR-216a-5p	Low levels associated with shorter overall survival and disease-free survival	Regulation of cell proliferation in vitro	HK2	[243]
miR-224	hsa-miR-224-5p	Associated with high risk			[238]
miR-224-5p		Downregulated in tumor tissues	Regulation of cell proliferation, migration and invasion in vitro	PIK3R3, AKT3	[244]
		Downregulated in tumor tissues	Regulation of proliferation and migration in vitro		[245]
miR-23a	hsa-miR-23a-3p		Regulation of cell migration and EMT in vitro	CDH1, VIM, ZEB1 (negative feedback loop with miR-23a)	[246]
miR-26a	hsa-miR-26a-5p		Regulation of cell viability, proliferation, and apoptosis in vitro	MDM2, TP53	[247]
		Downregulated in UM cell lines compared to normal choroidal cells	Regulation of cell proliferation in vitro	EZH2	[248]
miR-27a	hsa-miR-27a-3p	Downregulated after genistein administration in vitro	Regulation of cell proliferation in vitro	ZBTB10	[249]
miR-296-3p	hsa-miR-296-3p	Downregulated in choroidal malignant melanoma cell lines compared to normal melanocytes	Regulation of cell proliferation, migration, invasion, and apoptosis in vitro	MMP2, MMP9	[250]
miR-346	hsa-miR-346	Upregulated in metastatic tumors			[220]
miR-34a	hsa-miR-34a-5p	Downregulated in tumor tissues	Regulation of cell proliferation and migration in vitro	MET	[251]
			Regulation of cell migration, invasion and EMT in vitro	LGR4	[252]
		Upregulated in tumor tissues			[231]
miR-34b	hsa-miR-34b-5p	Downregulated in tumor tissues; downregulated after doxorubicin administration in vitro	Regulation of cell proliferation and migration in vitro; involved in sensitivity to doxorubicin	MET	[253]
miR-34c	hsa-miR-34c-5p	Downregulated in tumor tissues; downregulated after doxorubicin administration	Regulation of cell proliferation and migration in vitro; involved in sensitivity to doxorubicin	MET	[253]
miR-365a	hsa-miR-365a-3p	Associated with high risk			[238]
miR-365b	NA	Associated with high risk			[238]
miR-367	hsa-miR-367-3p	Upregulated in tumor tissues	Regulation of cell proliferation and migration in vitro	PTEN	[254]
miR-372	NA			SRC3/NCOA3	[217]
miR-378*	hsa-miR-378a-5p	Downregulated in tumors with chromosome 3 monosomy			[221]
miR-378d	hsa-miR-378d	Downregulated in high-risk tumors			[202]
		Upregulated in tumor tissues			[228]
miR-378g	hsa-miR-378g	Upregulated in tumor tissues			[228]
miR-429	hsa-miR-429		Regulation of cell proliferation in vitro	SRC3/NCOA3	[217]
miR-449b	hsa-miR-449b-5p	Upregulated in tumors with chromosome 3 monosomy			[221]
miR-452	hsa-miR-452-5p	Associated with high risk			[238]
miR-4709	NA	Associated with high risk			[238]
miR-506	hsa-miR-506-3p	Downregulated in tumors with chromosome 3 monosomy; upregulated in tumors with chromosome 3 monosomy and metastatic disease			[221]
		Downregulated in metastatic tumors			[220]
miR-506-3p		Downregulated in metastatic tumors			[240]
		High levels associated with better survival outcomes			[216]
miR-507	hsa-miR-507	Downregulated in tumors with chromosome 3 monosomy			[221]
		Low expression associated with poor prognosis			[235]
miR-508	hsa-miR-508-3p	Downregulated in advanced tumors			[222]
miR-508-3p		Downregulated in tumors with chromosome 3 monosomy; upregulated in tumors with chromosome 3 monosomy and metastatic disease			[221]
miR-508-3p		Downregulated in high- vs. low-grade tumors and deceased vs. alive patients; associated with poor prognosis and worse overall survival			[239]
		Downregulated in metastatic tumors			[240]
		High levels associated with better survival outcomes			[216]
miR-508-5p	hsa-miR-508-5p	Downregulated in metastatic tumors			[240]
miR-509-3-5p	hsa-miR-509-3-5p	Downregulated in tumors with chromosome 3 monosomy; upregulated in tumors with chromosome 3 monosomy and metastatic disease			[221]
		Downregulated in high- vs. low-grade tumors and deceased vs. alive patients; associated with poor prognosis and worse overall survival			[239]
		Downregulated in metastatic tumors			[240]
miR-509-3p	hsa-miR-509-3p	Downregulated in tumors with chromosome 3 monosomy; upregulated in tumors with chromosome 3 monosomy and metastatic disease			[221]
		Downregulated in metastatic tumors			[240]
miR-513a-5p	hsa-miR-513a-5p	Downregulated in tumors with chromosome 3 monosomy; upregulated in tumors with chromosome 3 monosomy and metastatic disease			[221]
		Downregulated in high- vs. low-grade tumors and deceased vs. alive patients; associated with poor prognosis and worse overall survival			[239]
		Downregulated in metastatic tumors			[240]
miR-513a-5p	hsa-miR-513a-5p	High levels associated with better survival outcomes			[216]
miR-513b	NA	Downregulated in tumors with chromosome 3 monosomy; upregulated in tumors with chromosome 3 monosomy and metastatic disease			[221]
miR-513b-5p	hsa-miR-513b-5p	Downregulated in metastatic tumors			[240]
miR-513c	hsa-miR-513c-5p	Protective miRNA			[238]
		Downregulated in metastatic tumors			[220]
miR-513c-5p		Downregulated in high- vs. low-grade tumors and deceased vs. alive patients; associated with poor prognosis and worse overall survival			[239]
		Downregulated in metastatic tumors			[240]
miR-514	hsa-miR-514a-3p	downregulated in tumors with chromosome 3 monosomy			[221]
miR-514a-3p		Downregulated in high- vs. low-grade tumors and deceased vs. alive patients; associated with poor prognosis and worse overall survival			[239]
		Downregulated in metastatic tumors			[240]
miR-519d	NA		Regulation of cell proliferation in vitro	SRC3/NCOA3	[217]
miR-548	NA	Low expression associated with poor prognosis			[235]
miR-592	hsa-miR-592	Downregulated in tumors with chromosome 3 monosomy			[221]
		Upregulated in high- vs. low-grade tumors and deceased vs. alive patients; associated with poor prognosis and worse overall survival			[239]
		Upregulated in metastatic tumors; correlated with chromosome 3 status; upregulated in monosomic tumors with or without loss of BAP1 expression			[220]
		Upregulated in metastatic tumors			[240]
miR-624	hsa-miR-624-3p	Upregulated in tumors with chromosome 3 monosomy; upregulated in tumors with chromosome 3 monosomy and metastatic disease			[221]
miR-624*	hsa-miR-624-5p	Downregulated in tumors with chromosome 3 monosomy			[221]
miR-628-5p	hsa-miR-628-5p	Upregulated in tumors with chromosome 3 monosomy			[221]
miR-651	NA	Downregulated in tumors with chromosome 3 monosomy			[221]
miR-652	hsa-miR-652-3p	Upregulated in high-risk tumors			[201]
		Upregulated in tumor tissues	Regulation of cell proliferation and migration in vitro	HOXA9	[255]
miR-708-5p	hsa-miR-708-5p	Upregulated in metastatic tumors			[240]
miR-7702	hsa-miR-7702	Associated with high risk			[238]
miR-873	hsa-miR-873-5p	Protective miRNA			[238]
miR-876-3p	hsa-miR-876-3p	Downregulated in tumors with chromosome 3 monosomy			[221]
miR-9	hsa-miR-9-5p	Negatively correlated with invasion of UM cells	Regulation of migration and invasion in vitro	NFKB1 (which targeted MMP2, MMP9, and VEGFA)	[256]
miR-92a-3p	hsa-miR-92a-3p	Downregulated after treatment with histone deacetylase inhibitor MS-275; upregulated in UM tissues compared to normal skin	Regulation of apoptosis in vitro	MYCBP2	[257]
miR-92b	hsa-miR-92b-3p	Upregulated in tumors with chromosome 3 monosomy			[221]
miR-935	hsa-miR-935	Downregulated in tumors with chromosome 3 monosomy; upregulated in tumors with chromosome 3 monosomy and metastatic disease			[221]
miR-93-5p	hsa-miR-93-5p			SRC3/NCOA3	[217]
miR-99a	hsa-miR-99a-5p	Downregulated in tumors with chromosome 3 monosomy			[221]
miR-99a-5p		Downregulated in high-risk tumors			[202]
miR-99a-3p	hsa-miR-99a-3p	Downregulated in high-risk tumors			[202]

**Table 3 cancers-15-00775-t003:** Dysregulated lncRNAs in UM.

lncRNA	Dysregulation	Function	Targets	Ref
*AC008555.4*	Low levels associated with poor prognosis			[268]
*AC016747.1*	Included in a prognostic signature; high levels associated with low survival rates			[269]
	Included in a prognostic signature; high levels associated with low survival rates			[270]
*AC016757.1*	Included in a prognostic signature; low levels associated with low survival rates			[269]
*AC018529.1*	High levels associated with poor prognosis			[268]
*AC018904.1*	Included in a prognostic signature; high levels associated with low survival rates			[270]
*AC090617.5*	Included in a prognostic signature; low levels associated with low survival rates			[270]
*AC100791.3*	Included in a prognostic signature; high levels associated with low survival rates			[270]
*AC104117.3*	Downregulated in high-risk tumors			[271]
*AC104129.1*	High levels associated with poor prognosis			[268]
	Downregulated in high-risk tumors			[272]
	Associated with high risk			[273]
*AC104825.1*	Included in a prognostic signature; low levels associated with low survival rates			[270]
*AC136475.3*	Downregulated in high-risk tumors			[272]
*ACVR2B-AS1*	Low levels associated with low survival rates			[274]
*AL589843.1*	Included in a prognostic signature; low levels associated with low survival rates			[269]
*ANRIL/CDKN2B-AS1*	Upregulated in tumor tissues	Regulation of INK4A/CDKN2A and INK4B/CDKN2B expression; regulation of metastasis in vitro and tumor growth in vivo		[275]
*AP005121.1*	Included in a prognostic signature; high levels associated with low survival rates			[269]
	Upregulated in high-risk tumors			[271]
*BACE1-AS*	Low levels associated with low survival rates			[276]
*BOLA3-AS1*	Included in a prognostic signature; low levels associated with low survival rates			[235]
*CASC15-NT1/CANT1*	Downregulated in tumor tissues	Regulation of cell migration and colony formation ability in vitro, and tumor growth in vivo	FTX; JPX; (XIST indirectly)	[277]
*circ_0032148*	Downregulated in tumor tissues		miR-181d-3p; miR-197-3p; miR-197-5p	[278]
*circ_0047924*	Upregulated in tumor tissues		miR-204-3p; miR-22-5p; miR-338-3p	[278]
*circ_0103232*	Upregulated in tumor tissues		miR-214-3p; miR-143-5p; miR-34a-3p	[278]
*circ_0119872*	Upregulated in tumor tissues	Regulation of cell proliferation, migration, invasion, and angiogenesis in vitro and in vivo	G3BP1 by sponging miR-622	[279]
*circ_0119873*	Upregulated in tumor tissues		miR-92a-3p; miR-193a-5p; miR-204-3p	[278]
*circ_0128533*	Upregulated in tumor tissues		miR-145-3p; miR-23a-5p; miR-23b-5p	[278]
*circ_0133460*	Downregulated in tumor tissues		let-7a-2-3p; let-7c-3p; miR-193a-5p	[278]
*circRNA10628-6*	Upregulated in tumor tissues		miR-197-5p; miR-214-3p; miR-34a-3p	[278]
*CYTOR*	High levels associated with poor prognosis			[268]
	High levels associated with low survival rates			[274]
*DKFZP434A062*	Low levels associated with low survival rates			[274]
*DLGAP1AS2*	Associated with high risk			[273]
*EPB41L4A-AS1*	Included in a prognostic signature; low levels associated with low survival rates			[235]
*FOXCUT1*	Downregulated in choroidal malignant melanoma cell lines compared to normal melanocytes	Regulation of cell proliferation, migration, invasion, and apoptosis in vitro	MMP2; MMP9	[250]
*FTH1P3*	Upregulated in tumor tissues	Regulation of cell proliferation and migration in vitro	FDZ5; RAC1; miR-224-5p (negative feedback loop with FTH1P3)	[244]
*GAS5*	Included in a prognostic signature			[235]
	Downregulated in tumor tissues; low levels associated with poor prognosis	Regulation of cell viability, migration, invasion, and EMT in vitro	miR-21-5p	[280]
*HCP5*	Included in a prognostic signature			[235]
*HOXA11-AS*	Upregulated in tumor tissues	Regulation of cell proliferation and apoptosis in vitro	p21/CDKN1A; EZH2; miR-124-3p	[218]
*IDI2-AS1*	Associated with low risk			[273]
*LINC00518*	Upregulated in tumor tissues; upregulated after triggering of EMT and hypoxia-like response; downregulated after MITF inhibition	Regulation of cell proliferation and migration in vitro	LINGO2; NFIA; OTUD7B; SEC22C; VAMP3	[281]
*LINC00634* */SMIM45*	Upregulated in tumor tissues			[281]
*LINC00957*	Included in a prognostic signature			[269]
*LINC00963*	Downregulated in high-risk tumors	Regulation of cell proliferation, migration, and invasion in vitro		[272]
*LINC01615*	High levels associated with low survival rates			[274]
*LINC02367*	Low levels associated with low survival rates			[274]
*LINC02572*	Low levels associated with low survival rates			[274]
*LOC100132707/PAXIP1-AS2*	Upregulated in metastatic tumors	Regulation of cell migration and invasion in vitro and tumor growth in vivo	JAK2	[282]
*LOC100505912/OUM1*	Upregulated in tumor tissues	Regulation of cell viability, proliferation, migration, and invasion in vitro, and tumor growth in vivo	PTPRZ1	[283]
*LOC101928143/lncRNA-numb*	Downregulated in UM cells compared to dermal fibroblasts; upregulated after HIC1 overexpression	Regulation of cell proliferation, colony formation and invasion in vitro		[284]
*MALAT1*	Upregulated in tumor tissues	Cell proliferation, colony formation, migration, and invasion in vitro	Ki-67/MKI67; PCNA; miR-140-3p	[227]
		Regulation of tumor growth in vivo	HOXC4 by sponging miR-608	[285]
*MIR4435-2HG*	High levels associated with poor prognosis			[268]
*P2RX7-V3/P2RX7 variant 3*	Upregulated in invasive UM cell lines compared to ARPE-19 cell line	Regulation of cell migration and colony formation ability in vitro, and tumor growth and progression in vitro	CDH1; Ki-67/MKI67; VIM	[286]
*PAUPAR*	Downregulated in tumor tissues	regulation of colony formation and migration in vitro, and tumor growth in vivo	HES1	[287]
*PPP1R14B-AS1*	Downregulated in high-risk tumors	Regulation of cell proliferation, migration, and invasion in vitro		[272]
*PVT1*	Included in a prognostic signature; high levels associated with low survival rates			[235]
	Upregulated in tumor tissues	Regulation of cell proliferation, colony formation ability and apoptosis in vitro	EZH2	[288]
	Upregulated in tumor tissues	Regulation of cell proliferation, migration, and invasion in vitro	MDM2 (and consequently TP53) by sponging miR-17-3p	[289]
	High levels associated with malignant features and poor overall survival; upregulation sustained by DNA amplification and methylation			[290]
	Included in a prognostic signature; high levels associated with low survival rates			[269]
*RHPN1-AS1*	Upregulated in tumor tissues	Regulation of cell proliferation, migration, and invasion in vitro and tumor growth in vivo		[291]
*RP11-329N22.1*	Upregulated in metastatic tumors			[292]
*RP1-272L16.1*	Upregulated in metastatic tumors			[292]
*SAMMSON*	Upregulated in metastatic tumors	Regulation of cell viability and apoptosis in vitro and tumor growth in vivo	p32/C1QBP; XRN2	[293]
*SNHG15*	High levels associated with worse prognosis, pathologic state, and metastasis			[294]
*SNHG7*	Included in a prognostic signature; low levels associated with low survival rates			[235]
	Low levels associated with poor prognosis; downregulated in metastatic tumors	Regulation of cell cycle arrest and apoptosis in vitro	EZH2	[295]
*SOS1-IT1*	Included in a prognostic signature; high levels associated with low survival rates			[270]
	Associated with high risk			[273]
*SOX1-OT*	Upregulated in high-risk tumors			[271]
*ZNF350-AS1*	Low levels associated with low survival rates			[274]
*ZNF667-AS1*	Downregulated in metastatic tumors; low levels associated with histological-type, metastasis, recurrence, death, and poorer prognosis	Regulation of cell viability, cell cycle progression and apoptosis	MEGF10	[296]
	Low levels associated with low survival rates			[274]
	Upregulated in high-risk tumors	Regulation of cell proliferation, migration, and invasion in vitro		[272]
*ZNNT1*	Downregulated in tumor tissues	Regulation of autophagy, cell death and migration in vitro, and tumor growth in vivo	ATG12	[297]

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
