# Peer review of "Genetics and RNA Regulation of Uveal Melanoma"

_cancers, 2023, doi:10.3390/cancers15030775_

Round 1

Reviewer 1 Report

The manuscript “Molecular Genetics of Uveal Melanoma” provides a timely and comprehensive overview over UM genetics with special focus on non-coding RNAs (miRNAs and lncRNAs). This should be acknowledged when choosing the title that should clearly indicate non coding RNAs as the main focus.

The review is exhaustive, well written and, with the exceptions mentioned below, well documented.

The authors should address the following points:

Row 23: point mutations, further in the text the actually mutated genes are cited, among which BAP1 that not necessarily shows point mutation. Hence, I suggest to omit “point” or to substitute it by “somatic”.

Row 84: “... than CM” should read “than in CM”.

Row 116/117: “To date, no therapy has been shown to be effective in treating metastatic disease in 116 UM patients [37].” This is not true since Tebentafusp has been approved.

Table 1: The table lists all mutations encountered without indicating the frequency of the single mutations that varies largely. This information can easily be found in the COSMIC database and I think, reporting this information here is redundant. The table should indicate the frequent oncogenic mutations and, for BAP1, different types of mutations that abolish BAP1 protein function. The table contains references indicated as PMID that should be numbered according to the reference list of the paper.

Row 236 and following:

The discussion of GNAQ and GNA11 should enclose recent work on the direct comparison of the effects of the genes on metastasis and interaction with other proteins.

Row 311 and following:

The discussion of SF3B1 mutations should also enclose evidence of SF3B1 mutations in hematological cancers where another amino acid in proximity to the one changed in UM is substituted. Work addressing the genes whose transcripts undergo aberrant splicing in SF3B1 mutated cases should be discussed.

Row 372: Mutations of MBD4 need further discussion since they confer a mutator phenotype and might correlate with response to immunotherapy.

Evidence of additional mutations in the calcium signaling pathway should also be cited.

The paper of Lalonde et al. Opthalmol. Sci. 2022 should be discussed and cited in relation to other cytogenetic aberrations.

Author Response

The manuscript “Molecular Genetics of Uveal Melanoma” provides a timely and comprehensive overview over UM genetics with special focus on non-coding RNAs (miRNAs and lncRNAs). This should be acknowledged when choosing the title that should clearly indicate non coding RNAs as the main focus. The review is exhaustive, well written and, with the exceptions mentioned below, well documented.

According to reviewer suggestion, the title has been modified in “Genetics and RNA regulation of uveal melanoma”.

The authors should address the following points:

  1. Row 23: point mutations, further in the text the actually mutated genes are cited, among which BAP1 that not necessarily shows point mutation. Hence, I suggest to omit “point” or to substitute it by “somatic”.

As suggested by the reviewer, we omitted “point” to refer to all mutations in general.

  1. Row 84: “... than CM” should read “than in CM”.

The text has been corrected.

  1. Row 116/117: “To date, no therapy has been shown to be effective in treating metastatic disease in 116 UM patients [37].” This is not true since Tebentafusp has been approved.

We corrected the sentence in the new version of the manuscript, adding the information about Tebentafusp.

  1. Table 1: The table lists all mutations encountered without indicating the frequency of the single mutations that varies largely. This information can easily be found in the COSMIC database and I think, reporting this information here is redundant. The table should indicate the frequent oncogenic mutations and, for BAP1, different types of mutations that abolish BAP1 protein function. The table contains references indicated as PMID that should be numbered according to the reference list of the paper.

According to reviewer suggestion, we highlighted in bold the most frequent mutations within the table. PMID IDs have been converted in citations of bibliography.

  1. Row 236 and following:

The discussion of GNAQ and GNA11 should enclose recent work on the direct comparison of the effects of the genes on metastasis and interaction with other proteins.

Following the appropriate suggestion of the reviewer, we added the following paragraph in the new version of the manuscript:

“Less is known about the prevalence and significance of GNAQ and GNA11 mutations in metastatic UM (MUM) [35804836]. The first studies on the role of these genes in UM prognosis reported that the distribution of GNA11 and GNAQ mutations varied between primary tumors and MUM, with a GNA11 to GNAQ ratio of 0.7 in primary UM and 2.6 in MUM [21083380, 35804836]. Griewank et al. discovered that GNA11 mutations were significantly more common than GNAQ mutations in metastatic specimens. Furthermore, patients with GNA11-mutant tumors had lower disease-specific survival and OS compared to wild-type patients. The authors proposed that the survival data, combined with the predominance of GNA11 mutations in metastasis, raises the possibility that GNA11-mutant tumors may be associated with a higher risk of metastasis and a worse prognosis than GNAQ-mutant tumors [23887304, 35804836].

Terai et al. recently investigated the existence of a relationship between metastasis-to-death and frequency of GNAQ and GNA11 mutations in eighty-seven MUM patients. The authors reported a similar mutation rate for GNA11 and GNAQ mutations (47.1% and 44.8% of patients, respectively) [34830903]. This result was consistent with previous findings for primary UM [19078957, 21083380]. Moreover, they discovered that differences in the type of mutation (p.Q209 vs. p.Q209L) rather than the GNAQ and GNA11 genes themselves could predict MUM patient survival [34830903, 35804836].

Functional differences between GNA11 and GNAQ might be determined by different interaction partners. To investigate this aspect, Piaggio at al. conducted a study using Tandem Affinity Purification and Mass Spectrometry (TAP-MS/MS) [35580369] to identify proteins that interact with GNAQ or GNA11. The comparison of the protein interaction networks of the two Gα-proteins only showed a very limited overlap, indicating functional differences between GNAQ and GNA11 [35580369]. The interaction of mutated GNAQ with the dioxygenase TET2 (tet methylcytosine dioxygenase 2), which is not observed for mutated GNA11, was confirmed by coimmunoprecipitation analyses. Interestingly, TET2 plays an active role in DNA demethylation and high-risk UMs are characterized by widespread demethylation [35580369].”

  1. Row 311 and following:

The discussion of SF3B1 mutations should also enclose evidence of SF3B1 mutations in hematological cancers where another amino acid in proximity to the one changed in UM is substituted. Work addressing the genes whose transcripts undergo aberrant splicing in SF3B1 mutated cases should be discussed.

Following the suggestion of the reviewer, we added these parts of discussion in the new version of the manuscript:

“SF3B1 mutations are mostly identified in hematolymphoid malignancies. Mutations in codon 700 represent 50% of all of the reported alterations; other mutations were found in codons 666, 662, 622, and 625 [36230848].

In UM, SF3B1 mutations almost exclusively occur in codon 625 and have been identified in 4% to 24% of primary tumors [26923342, 32340176].”

“By using RNA-Seq analyses of UM, Alsafadi et al., showed that the SF3B1 mutations resulted in deregulated splicing at a subset of splice junctions, mostly by the use of alternative 3′ acceptor splice site (3′ss) [26842708]. At first, they observed that SF3B1 hotspot mutations in UM were associated with the deregulation of a restricted subset (∼0.5%) of splice junctions, mostly caused by the usage of alternative 3′ss (AG′) upstream of the canonical 3′ss (AG). Second, they showed that splicing alterations induced by SF3B1 mutations were not reproduced either by knockdown or by overexpressing wild-type SF3B1, indicating that SF3B1 mutants may be qualified as change-of-function mutants. Third, their results provided significant progresses in understanding the molecular mechanisms underlying alternative 3′ss regulation by mutated SF3B1 [26842708]. Such mechanism involves a misregulation of branchpoint (BP) usage, which has been largely overlooked in previous studies on alternative splicing [25561518, 26842708].”

  1. Row 372: Mutations of MBD4 need further discussion since they confer a mutator phenotype and might correlate with response to immunotherapy.

Following the suggestion of the reviewer, we added this part of discussion on MBD4 in the new version of the manuscript:

“MBD4 (methyl-CpG binding domain 4, DNA glycosylase), mapping on chromosome 3, encodes for a DNA glycosylase involved in the repair of C>T mutations arising from spontaneous deamination of 5-methylcytosine. Given the high frequency of chromosome 3 monosomy in UM (see below), MBD4 is often present in single copy; in this scenario, a single mutation is sufficient for the inactivation of MBD4. Hence, MBD4 may act as tumor suppressor in UM [31277343, 32239153]. Accordingly, Darrien et al. reported that some UMs display a high level of CpG>TpG mutations in association with mutational inactivation of MBD4. In particular, germline protein truncating variants (PTVs) and somatic loss of the wild-type allele were reported in UM patients with a CpG>TpG mutator phenotype. MBD4 was suggested as a new predisposing gene for UM; indeed, it was associated with hypermutated M3 tumors and conferred predisposition to high-risk tumors [32239153]. Recently, MBD4 was reported as a prognostic factor for response to immune checkpoint inhibitors in MUM patients [35863105].”

  1. Evidence of additional mutations in the calcium signaling pathway should also be cited.

Following the suggestion of the reviewer, we added this part of discussion in the new version of the manuscript:

“Another alteration caused by GNA11 and GNAQ mutations concerns the calcium signaling pathway, whose dysregulation has a well-documented association with cancer survival, proliferation, migration, and metastatic potential [31545410]. For example, calcium signaling has been reported to be involved in the proliferation of RAS-driven cancers through the interaction between calmodulin and PI3K (phosphatidylinositol-4,5-bisphosphate 3-kinase) [28462395, 31545410] and the promotion of invasion and metastasis via ERK activation in both BRAF-driven and non-BRAF CM cells [24586666, 31545410]. Constitutive activation of Gαq signaling by mutations in GNAQ or GNA11 occurs in over 80% of UMs and activates MAPK signaling [28486107]. Chen et al. reported that RAS oncoproteins are required for GNAQ-mediated MAPK activation and identified PRKCD (protein kinase C delta), PRKCE (protein kinase C epsilon) and RASGRP3 (RAS guanyl releasing protein 3) as components of a signaling module necessary and sufficient to activate the Ras/MAPK pathway in GNAQ mutant UM. RASGRP3 is selectively overexpressed in response to GNAQ/GNA11 mutations in UM; its activation occurs via PRKCD- and PRKCE-dependent phosphorylation and PKC (protein kinase C)-independent, DAG (diacylglycerol)-mediated membrane recruitment, possibly explaining the limited effect of PKC inhibitors in durably suppressing MAPK in UM. The results achieved by Chen et al. suggested RASGRP3 as a therapeutic target for cancers driven by oncogenic GNAQ/GNA11 [28486107].”

  1. The paper of Lalonde et al. Opthalmol. Sci. 2022 should be discussed and cited in relation to other cytogenetic aberrations.

Following the suggestion of the reviewer, we added a brief discussion about other cytogenetic aberrations in the new version of the manuscript:

“Lalonde et al. evaluated the clinical relevance of low-frequency copy number aberrations (CNAs) in UM. Their study, based on genomic profiling of 921 primary tumors, revealed CNAs associated with the risk of metastasis and demonstrated a strong association between chromosomal instability and patient prognosis. Their results suggested that 1p and 16q deletions should be incorporated in clinical assays to assess prognosis at diagnosis and to guide enrollment in clinical trials for adjuvant therapies [36249692]”.

Reviewer 2 Report

The manuscript by Barbagallo et al., titled "Molecular Genetics of Uveal Melanoma" is a comprehensive review article that incorporates all of the most up-to-date findings regarding the miRNA, LncRNA, and cirRNA. The author exercised excellent discretion in writing the manuscript. Every possible treatment option for uveal melanoma, both new and old, is covered in detail here. The review article's authors covered both old and contemporary studies of microRNAs, long noncoding RNAs (lncRNAs), and circular RNAs (cirRNAs). The literature surveying this subject is scant. As a result, the reviewed article in its current form is acceptable.

Author Response

The manuscript by Barbagallo et al., titled "Molecular Genetics of Uveal Melanoma" is a comprehensive review article that incorporates all of the most up-to-date findings regarding the miRNA, LncRNA, and cirRNA. The author exercised excellent discretion in writing the manuscript. Every possible treatment option for uveal melanoma, both new and old, is covered in detail here. The review article's authors covered both old and contemporary studies of microRNAs, long noncoding RNAs (lncRNAs), and circular RNAs (cirRNAs). The literature surveying this subject is scant. As a result, the reviewed article in its current form is acceptable.

We thank the reviewer for their time and expertise, and appreciate their supportive comments. We thank the reviewer for its time and evaluation of our manuscript.

Round 2

Reviewer 1 Report

The authors have adequately addressed all issues raised.